# DyME: Dynamic Multi-Concept Erasure in Diffusion Models with Bi-level Orthogonal LoRA Adaptation

## Abstract

Text-to-image diffusion models (DMs) inadvertently reproduce copyrighted styles and protected visual concepts, raising legal and ethical concerns. Concept erasure has emerged as a safeguard, aiming to selectively suppress such concepts through fine-tuning. However, existing methods do not scale to practical settings where providers must erase multiple and possibly conflicting concepts. The core bottleneck is their reliance on static erasure: a single checkpoint is fine-tuned to remove all target concepts, regardless of the actual erasure needs at inference. This rigid design mismatches real-world usage, where requests vary per generation, leading to degraded erasure success and reduced fidelity for non-target content.

We propose DyME, an on-demand erasure framework that trains lightweight, concept-specific LoRA adapters and dynamically composes only those needed at inference. This modular design enables flexible multi-concept erasure, but naive composition causes interference among adapters, especially when many or semantically related concepts are suppressed. To overcome this, we introduce bi-level orthogonality constraints at both the feature and parameter levels, disentangling representation shifts and enforcing orthogonal adapter subspaces. We further develop ErasureBench-H, a new hierarchical benchmark with brand–series–character structure, enabling principled evaluation across semantic granularities and erasure set sizes. Experiments on ErasureBench-H and standard datasets (*e.g.*, CIFAR-100, Imagenette) demonstrate that DyME consistently outperforms state-of-the-art baselines, achieving higher multi-concept erasure fidelity with minimal collateral degradation.

## 1 Introduction

Recent advances in text-to-image diffusion models (DMs) (Rombach et al., 2022), have enabled remarkable generation capabilities across a vast range of visual concepts. This expressiveness, however, has raised pressing legal and ethical issues: DMs can easily reproduce copyrighted content such as trademarked characters, corporate logos, and proprietary designs (Jiang et al., 2023; Zhang et al., 2023a; Almeda et al., 2024), exposing providers to increasing legal scrutiny and lawsuits (Winston Cho, 2024; Chris Cooke, 2024). To mitigate these risks, *concept erasure* has emerged as a practical safeguard that disables a model's ability to generate protected or unwanted content while preserving quality for unrelated concepts. Typical methods fine-tune the DM so that prompts invoking a target concept (e.g., "a picture of Mickey Mouse") are redirected to neutral substitutes (e.g., "a generic cartoon character"), enabling targeted removal without retraining from scratch (Zhang et al., 2024a; Lyu et al., 2024; Orgad et al., 2023; Gandikota et al., 2024; Gong et al., 2024; Fan et al., 2024).

While effective in the single-concept case, existing methods struggle with the multi-concept erasure required in practice, such as takedown requests covering all copyrighted characters in a specific series, brand or arbitrary combinations thereof. As the *erasure scope* (the full set of concepts prepared for removal) expands, their performance deteriorates due to two core limitations. First, parameter-level conflicts arise when model updates for multiple concepts, causing gradients to clash and weakening the removal of a large number of concepts. Second, at the semantic level, related concepts often share latent attributes or representation directions, making selective suppression difficult. This leads to both leakage of target concepts and degradation of generating benign content (Nie et al., 2025; Kumari et al., 2023).

A closer look reveals that these failures stem from the *static erasure paradigm* adopted by prior methods. Each fine-tuning run targets a fixed set of concepts, producing a checkpoint that can only suppress this set as a whole, regardless of the per-generation erasure subset (the specific set requested at inference). This rigid design mismatches real-world demands, where erasure requests vary per generation and typically involve only the concepts explicitly invoked in a user's prompt. For example, if the erasure scope covers all Disney characters, static methods suppress every character in that set for every prompt. Yet in practice, if a user enters the prompt "a photo of Mickey Mouse" to generate an image, the erasure subset contains only Mickey Mouse, since no other Disney characters are involved. Static approaches cannot make this distinction: once trained on a broad erasure scope, they erase all included concepts indiscriminately, reducing diversity and degrading fidelity. As shown in Fig. 1 (Scenario 1), this static design scales poorly as the erasure scope widens.

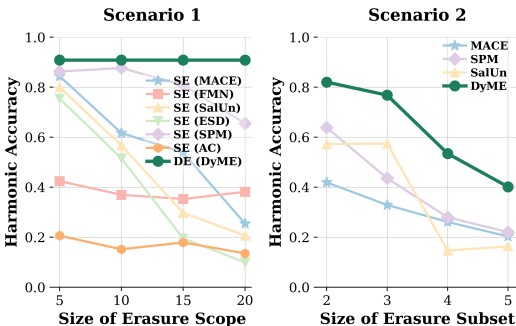

Figure 1: **Multi-Concept Erasure Scalability.** Scenario 1 (left): We increase erasure scope but in each generation only one concept is erased. Static erasure methods (**SE**: MACE, FMN, SalUn, ESD, SPM, AC) significantly degrade, whereas our dynamic erasure (**DE**: DYME) remains effective. Scenario 2 (right): We increase the number of concepts per generation (*erasure subset*), with fixed erasure scope, DYME significantly outperforms existing methods (MACE, SPM, SalUn). Harmonic accuracy is defined in Sec. 5.1.

To overcome these limitations, we shift to an *on-demand erasure framework*. The key idea is to decouple training from inference by distinguishing between the erasure scope and the erasure subset. This separation enables a modular design in which erasure components are trained collectively for coverage but activated selectively per generation. As a result, each request suppresses the necessary concepts, minimizing collateral damage and preserving generation quality for non-target content.

Building on this principle, we introduce DYME, a **dy**namic **m**ulti-concept **e**rasure framework that treats concept removal as an on-demand capability. DYME trains lightweight, concept-specific LoRA (Hu et al., 2022) adapters, and at inference, dynamically composes only those adapters corresponding to the erasure subset. This design provides per-generation control: when the erasure subset is fixed, performance remains stable as the erasure scope grows, as shown by the green curve in Scenario 1 of Fig. 1, yielding an unchanged and substantial advantage of DYME over static erasure methods. A key challenge, however, is LoRA crosstalk (Dalva et al., 2025; Gu et al., 2023; Po et al., 2024; Simsar et al., 2025): non-orthogonal updates from multiple adapters can interfere in shared layers especially cross-attention, degrading both erasure reliability and generation fidelity. To overcome this, we develop bi-level orthogonality constraints: an input-aware constraint that disentangles LoRA-induced representation shifts for specific prompts, and a parameter-level constraint that enforces independence across adapter weights globally. Together, these constraints ensure that adapters operate in complementary subspaces, enabling robust multi-concept erasure.

Finally, to enable rigorous evaluation of multi-concept erasure scalability, we extend prior evaluation that vary only the erasure scope by additionally scaling with the per-generation erasure subset. Concretely, we instantiate scaling erasure subset requests in two ways: (i) simply using conjunctions explicitly invoke multiple concepts per generation; and (ii) by enlarging the *concept scope* of a named concept, where concept scope is defined as the number of defined unit concepts it subsumes. However, on standard flat-category benchmarks such as CIFAR-100 (Krizhevsky et al., 2009) and Imagenette (Howard & Gugger, 2020), concepts are not hierarchically nested, making unit concepts and thus concept scope ill-defined. So we introduce ERASUREBENCH-H, a benchmark with a hierarchical *brand–series–character* structure that mirrors real-world takedown requests targeting groups of related concepts. This hierarchy makes concept scope explicit (e.g., a brand covers multiple series, which in turn cover multiple characters), thereby supporting controlled analyses across per-generation erasure subset sizes by varying concept scope. ERASUREBENCH-H thus provides a principled testbed for evaluating scalable, dynamic erasure methods beyond what flat-category datasets allow. To the best of our knowledge, this is the first work to systematically investigate multi-concept erasure scalability in diffusion models.

Our main contributions are threefold:

- We formalize multi-concept erasure in diffusion models, identify parameter- and semantic-level coupling as key barriers, and introduce the scope–subset distinction to enable scalable erasure.
- We propose DYME, a dynamic erasure framework that trains modular LoRA adapters and introduces bi-level orthogonality constraints to mitigate crosstalk, ensuring reliable multi-concept composition.
- We release ERASUREBENCH-H, a hierarchical benchmark for real-world multi-concept evaluation, and show through extensive experiments on CIFAR-100, Imagenette, and ERASUREBENCH-H that DYME consistently outperforms static baselines, achieving >90% harmonic accuracy even as the erasure scope grows. Moreover, when the size of erasure subset increases, DYME maintains a clear lead over all baselines.

## 2 RELATED WORK

**Concept erasure in diffusion models.** Recent work on concept erasure aims to remove targeted concepts from text-to-image diffusion models (e.g., Stable Diffusion) while preserving non-target fidelity (Fan et al., 2024; Li et al., 2024; Schramowski et al., 2023). Among fine-tuning approaches, FMN (Zhang et al., 2024a) suppresses targets by re-steering cross-attention (CA) scores of the corresponding tokens; ESD (Gandikota et al., 2023) aligns target concepts toward a surrogate distribution via CA-layer fine-tuning; and SPM (Lyu et al., 2024) inserts rank-1 parameter fine-tuning into selected layers, which are trained to map the target concept to a safe surrogate. UCE (Gandikota et al., 2024) provide closed-form updates for the cross-attention projection matrices, and yield fast edits. Besides, SalUn (Fan et al., 2024) uses a gradient-based saliency mask to update parameters most salient to the forgetting objective. MACE (Lu et al., 2024) couples a closed-form initialization with lightweight LoRA refinement, fusing per-concept modules together. However, by assuming a static erasure paradigm, prior work risks growing cross-concept interference. We investigate a dynamic framework to improve the scalability and reliability of multi-concept erasure.

**LoRA composition and interference mitigation.** LoRA (Hu et al., 2022) adapts diffusion models by injecting low-rank updates into linear layers while freezing base weights, enabling parameter-efficient personalization (Tewel et al., 2024). To preserve plug-and-play control at inference, composition techniques determine how multiple adapters interact. LoRA-Merge (Zhong et al., 2024) linearly fuses several low-rank deltas into the base weights to produce a single set of weights. LoRA-Switch (Zhong et al., 2024) keeps adapters separate and activates one adapter (or schedules different ones) across denoising steps. LoRA-Composite (Zhong et al., 2024) mixes multiple adapters via uniform or weighted averaging to support multi-concept/style control. However, simple composition can induce concept conflicts and identity loss (Gu et al., 2023). To address this, Mix-of-Show (Gu et al., 2023) formulates a constrained optimization to merge individually trained LoRAs while preserving identity; yet it ultimately consolidates multiple adapters into a single LoRA, reverting to a static erasure paradigm. Orthogonal Adaptation (Po et al., 2024) encourages zero inner products between per-concept parameter matrices, but abstracts away from analyzing the cross-attention projections where LoRA is actually injected; we address these gaps by proposing bi-level orthogonality constraints directly on these projections to better reduce interference, and adopting the training-free LoRA-Composite that enables dynamic erasure without retraining or per-subset checkpoints.

## 3 PROBLEM STATEMENT AND CHALLENGES

**Problem statement.** Concept erasure aims to disable a model's ability to generate specific visual concepts, such as copyrighted characters. Formally, let $\mathcal{C}$ denote the universe of possible visual concepts. An erasure scope $\mathcal{C}_{\text{scope}} \subseteq \mathcal{C}$ is specified as all concepts the model should be prepared to erase. At inference, a narrower erasure subset $\mathcal{C}_{\text{subset}} \subseteq \mathcal{C}_{\text{scope}}$ is identified, corresponding to the concepts that should be suppressed for a given prompt or generation. The goal of concept erasure is twofold: (1) For any prompt $p$ that invokes a target concept $c \in \mathcal{C}_{\text{subset}}$, the generated image $x_0$ should omit that concept; (2) For prompts containing only non-target concepts, $x_0$ should remain consistent with the original model's distribution.

**Challenges in current concept erasure methods.** While existing concept erasure methods primarily address the single-concept case, providers often face requests to suppress multiple related concepts, such as all copyrighted characters from a specific series or brand. This multi-concept erasure setting, where the model must handle arbitrary subsets $\mathcal{C}_{\text{subset}} \subseteq \mathcal{C}_{\text{scope}}$, introduces the risk of interference between concept updates.

A straightforward strategy is to fine-tune the model to suppress the entire $\mathcal{C}_{\text{scope}}$, a static approach. As the size of $\mathcal{C}_{\text{scope}}$ increases, interference accumulates between erased concepts, as well as between erased and preserved concepts. This results in degraded erasure effectiveness and lower non-target fidelity, making static erasure unsuited to dynamic or large-scale policies.

A more flexible strategy is to train concept-specific LoRA adapters and activate only those required for a particular $\mathcal{C}_{\text{subset}}$. This modular approach avoids unnecessary interference from irrelevant concepts and allows for dynamic erasure. However, when multiple adapters are activated together, their parameter updates can overlap in the shared model layers, causing destructive crosstalk. This is especially severe when the erased concepts are semantically similar or share visual features, leading to both erasure leakage and collateral degradation. Figure 2 illustrates this crosstalk, with heatmaps showing that LoRA-induced changes for similar concepts tend to align strongly (low orthogonality). Together, these challenges highlight the need for a principled framework that can support arbitrary erasure subsets while minimizing destructive crosstalk and preserving fidelity.

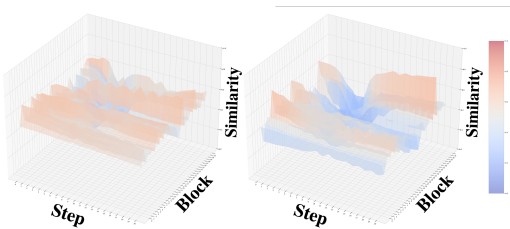

(a) Semantically similar LoRA pair    (b) Semantically dissimilar LoRA pair

Figure 2: **LoRA crosstalk analysis**. Cosine-similarity heatmaps of LoRA-induced changes in cross-attention outputs across timesteps and U-Net blocks. (a) Semantically similar pairs show high similarity (red), indicating overlapping updates and strong crosstalk. (b) Semantically dissimilar pairs are more orthogonal (blue), showing reduced interference.

## 4 DYME: DYNAMIC MULTI-CONCEPT ERASURE FRAMEWORK

To overcome the limitations of prior methods, we introduce DYME, a *Dynamic Multi-Concept Erasure* framework that treats concept erasure as an on-demand capability rather than a one-time fine-tune. Instead of producing a single static checkpoint tied to a fixed erasure scope, DYME equips a pre-trained DM with a set of lightweight, concept-specific LoRA modules. At inference, only the LoRAs corresponding to the requested erasure subset are activated and composed, enabling efficient and flexible erasure across arbitrary combinations without retraining or checkpoint management.

Figure 3 illustrates the DYME **workflow** in four steps. *Step 1*: Define the erasure scope $\mathcal{C}_{\text{scope}}$, the full set of concepts that may be erased, and specify neutral substitutes that determine how erased concepts should appear (e.g., background, empty, or generic replacements). In our setup, we adopt the absence variant as the reconstruction target. *Step 2*: Attach a lightweight LoRA module to the backbone for each concept $c_i \in \mathcal{C}_{\text{scope}}$. *Step 3*: Train all LoRA modules with a joint objective that combines reconstruction fidelity with orthogonality-based disentanglement, ensuring each module suppresses its target concept without interfering with others. *Step 4*: At inference, given a prompt and user-specified erasure subset $\mathcal{C}_{\text{subset}} \subseteq \mathcal{C}_{\text{scope}}$, DYME activates only the relevant LoRAs, composes their outputs into a single denoising direction, and generates the final image. This modular pipeline decouples training from inference: LoRAs are trained collectively for coverage but designed for composability, enabling scalable, per-demand erasure. We next describe how DYME enforces this composability through bi-level orthogonality constraints.

### 4.1 TRAINING WITH BI-LEVEL ORTHOGONALITY CONSTRAINTS

Naively composing multiple LoRAs for concepts in $\mathcal{C}_{\text{subset}}$ leads to crosstalk, particularly when concepts are semantically related and their induced updates overlap in shared layers. To enable stable multi-concept composition, we introduce a *bi-level orthogonality strategy* that integrates both input-aware and input-agnostic constraints, mitigating interference both on observed training data and in unseen scenarios.

**Input-aware orthogonality constraint.** To address prompt-specific interference during training, we regularize LoRAs to produce disentangled representation shifts in cross-attention activations. For each pair of concepts, we compute the change induced by their respective LoRA modules and penalize alignment between these shifts. This encourages LoRAs to act along orthogonal directions in the representation space, ensuring that composing adapters for the concepts and prompts seen during training does not introduce redundant or conflicting updates.

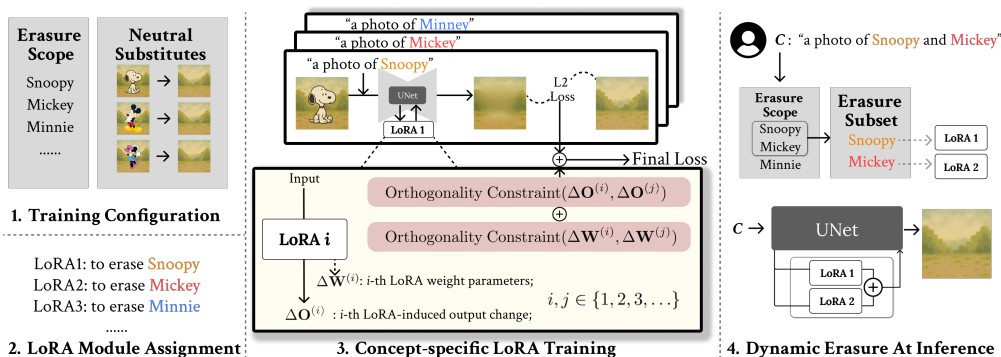

Figure 3: **DYME overview.** Workflow from scope definition and LoRA assignment to training with orthogonality constraints and dynamic composition at inference.

Suppose for concept $c_i$, the LoRA-induced update to the backbone weight matrices $\mathbf{W}_\star^{(0)}$ is $\Delta\mathbf{W}_\star^{(i)}$ for $\star \in \{q, k, v, o\}$ ($q, k, v, o$ denote, respectively, the query, key, value, and output projections of cross-attention). Let the modified weights be $\boldsymbol{W}_\star^{(i)} = \boldsymbol{W}_\star^{(0)} + \Delta\boldsymbol{W}_\star^{(i)}$, and let $\boldsymbol{X} \in \mathbb{R}^{d \times d_e}$ be the text embedding. The LoRA-modified output is:

$$\boldsymbol{O}^{(i)} = \boldsymbol{W}_o^{(i)}\boldsymbol{W}_v^{(i)}\boldsymbol{X} \cdot \sigma\left(\frac{(\boldsymbol{W}_k^{(i)}\boldsymbol{X})^\top \boldsymbol{W}_q^{(i)}\boldsymbol{z}^{(i)}}{\sqrt{d_e}}\right), \tag{1}$$

where $\boldsymbol{z}^{(i)}$ is the visual query token and $\sigma(\cdot)$ is the softmax. The induced shift is $\Delta\boldsymbol{O}^{(i)} = \boldsymbol{O}^{(i)} - \boldsymbol{O}^{(0)}$. We define the orthogonality score between two LoRA modules as:

$$\mathrm{OS}(i, j) = 1 - \frac{\langle\Delta\boldsymbol{O}^{(i)}, \Delta\boldsymbol{O}^{(j)}\rangle_F}{\|\Delta\boldsymbol{O}^{(i)}\|_F \|\Delta\boldsymbol{O}^{(j)}\|_F}, \tag{2}$$

where $\langle\cdot, \cdot\rangle_F$ denotes the Frobenius inner product. Then we construct the input-aware orthogonality loss as:

$$\mathcal{L}_{\mathrm{ortho}}^{\mathrm{aware}} = -\mathbb{E}_{(c_i, c_j) \sim \mathcal{C}_{\mathrm{scope}}, i \neq j}\left[\mathrm{OS}(i, j)\right], \tag{3}$$

which penalizes correlated LoRA-induced shifts across pairs, sampled from the erasure scope.

**Input-agnostic orthogonality constraints.** While effective on observed data, input-aware constraints alone can be limited: they depend on the coverage of training prompts and may leave residual overlap in unobserved or biased input distributions. In other words, orthogonality enforced on training samples does not guarantee global disentanglement, especially as real-world prompts are diverse and unpredictable.

To address these gaps, we introduce an input-agnostic constraint, which operates directly in the parameter space of the LoRA modules, independent of specific input prompts. By encouraging the parameters associated with each concept's LoRA to be orthogonal, we promote global disentanglement and robustness to prompt distribution shift. To make this precise, we formalize a parameter-space condition that serves as an input-agnostic surrogate for output orthogonality:

**Theorem 1.** *Let $c_i$ and $c_j$ be any two distinct concepts ($i \neq j$). Suppose LoRA adaptation is restricted to the value and output projections $\boldsymbol{W}_v$, $\boldsymbol{W}_o$, with the query and key projections fixed ($\boldsymbol{W}_q^{(i)} = \boldsymbol{W}_q^{(0)}$, $\boldsymbol{W}_k^{(i)} = \boldsymbol{W}_k^{(0)}$). Define*

$$\boldsymbol{M}^{(i)} := \boldsymbol{W}_o^{(0)}\Delta\boldsymbol{W}_v^{(i)} + \Delta\boldsymbol{W}_o^{(i)}\boldsymbol{W}_v^{(0)} + \Delta\boldsymbol{W}_o^{(i)}\Delta\boldsymbol{W}_v^{(i)}.$$

*Then a sufficient condition for the orthogonality of the representation shifts, $\langle\Delta\boldsymbol{O}^{(i)}, \Delta\boldsymbol{O}^{(j)}\rangle_F = 0$, for all input text embeddings $\boldsymbol{X}$ and queries $\boldsymbol{z}^{(i)}, \boldsymbol{z}^{(j)}$, is that*

$$(\boldsymbol{M}^{(i)})^\top\boldsymbol{M}^{(j)} + (\boldsymbol{M}^{(j)})^\top\boldsymbol{M}^{(i)} = \boldsymbol{0}.$$

*Proof.* Under the stated assumptions, the LoRA-induced representation shift for concept $c_i$ is

$$\Delta\boldsymbol{O}^{(i)} = \left(\boldsymbol{W}_o^{(i)}\boldsymbol{W}_v^{(i)} - \boldsymbol{W}_o^{(0)}\boldsymbol{W}_v^{(0)}\right)\boldsymbol{X}\boldsymbol{A}^{(i)}$$

$$= \left((\boldsymbol{W}_o^{(0)} + \Delta\boldsymbol{W}_o^{(i)})(\boldsymbol{W}_v^{(0)} + \Delta\boldsymbol{W}_v^{(i)}) - \boldsymbol{W}_o^{(0)}\boldsymbol{W}_v^{(0)}\right)\boldsymbol{X}\boldsymbol{A}^{(i)}$$

$$= \boldsymbol{M}_i\boldsymbol{X}\boldsymbol{A}^{(i)},$$

where

$$A^{(i)} = \sigma \left( \frac{(W_k^{(0)} X)^\mathsf{T} W_q^{(0)} z^{(i)}}{\sqrt{d_e}} \right).$$

Since we assume the attention map $A^{(i)}$ remains the same ($W_q$ and $W_k$ fixed), only $W_v$ and $W_o$ differ between concepts, and thus the output change due to LoRA is linear in $M^{(i)}$. The Frobenius inner product between the shifts for concepts $i$ and $j$ is then

$$\langle \Delta O^{(i)}, \Delta O^{(j)} \rangle_F = \mathrm{tr} \left[ (\Delta O^{(i)})^\mathsf{T} \Delta O^{(j)} \right]$$
$$= \mathrm{tr} \left[ (A^{(i)})^\mathsf{T} X^\mathsf{T} (M^{(i)})^\mathsf{T} M^{(j)} X A^{(j)} \right].$$

To guarantee this vanishes for all possible choices of $X$, $A^{(i)}$, $A^{(j)}$, it is sufficient that $(M^{(i)})^\mathsf{T} M^{(j)}$ is skew-symmetric: $(M^{(i)})^\mathsf{T} M^{(j)} + (M^{(j)})^\mathsf{T} M^{(i)} = 0$. Indeed, for any vectors $u, v$,

$$u^\mathsf{T} (M^{(i)})^\mathsf{T} M^{(j)} v = -v^\mathsf{T} (M^{(i)})^\mathsf{T} M^{(j)} u.$$

By the properties of the trace and bilinearity, this implies

$$\mathrm{tr} \left[ u^\mathsf{T} (M^{(i)})^\mathsf{T} M^{(j)} v \right] = 0,$$

and so $\langle \Delta O^{(i)}, \Delta O^{(j)} \rangle_F = 0$ for all $i \neq j$.

Thus, the proposed parameter-space constraint is sufficient for input-agnostic orthogonality between LoRA-induced representation shifts. □

This theorem establishes that LoRA orthogonality can be enforced through a symmetric condition on their parameter matrices, removing the need for input-dependent Jacobians or forward-pass correlations. Building on this result, we construct the input-agnostic orthogonality loss as

$$\mathcal{L}_{\mathrm{ortho}}^{\mathrm{agnostic}} = -\mathbb{E}_{(c_i, c_j) \sim \mathcal{C}_{\mathrm{scope}}, i \neq j} \left[ \left\| \tfrac{1}{2} \left( (M^{(i)})^\mathsf{T} M^{(j)} + (M^{(j)})^\mathsf{T} M^{(i)} \right) \right\|_F^2 \right]. \tag{4}$$

This loss encourages concept-specific LoRA modules to reside in decorrelated parameter subspaces, providing a lightweight and input-independent safeguard against crosstalk that complements the input-aware constraint.

**Final Training Objective.** The overall training objective for DYME combines a reconstruction loss (erasing target concepts), the input-aware orthogonality loss (reducing input-specific crosstalk), and the input-agnostic orthogonality loss (providing global disentanglement):

$$\mathcal{L} = \mathcal{L}_{\mathrm{rec}} + \lambda_1 \mathcal{L}_{\mathrm{ortho}}^{\mathrm{aware}} + \lambda_2 \mathcal{L}_{\mathrm{ortho}}^{\mathrm{agnostic}}, \tag{5}$$

where $\lambda_1$ and $\lambda_2$ control the relative strength of the two constraints. Here, $\mathcal{L}_{\mathrm{rec}}$ is a distance between the generated image and its neutral substitute, ensuring erasing effectiveness of target concepts, $\mathcal{L}_{\mathrm{ortho}}^{\mathrm{aware}}$ mitigates sample-specific interference, and $\mathcal{L}_{\mathrm{ortho}}^{\mathrm{agnostic}}$ enforces global disentanglement. Together, they ensure LoRAs are effective individually and stable when composed.

### 4.2 DYNAMIC COMPOSITION AT INFERENCE

At inference, DYME identifies the erasure subset $\mathcal{C}_{\mathrm{subset}}$ for each prompt $p$. Only the corresponding LoRA modules are activated, and their classifier-free guidance (Ho & Salimans, 2022) predictions are computed and averaged to yield a unified denoising direction. This enables on-demand erasure of arbitrary concept subsets, without retraining or storing multiple static checkpoints. Crucially, thanks to bi-level orthogonality constraints, performance remains stable as the erasure scope grows.

## 5 EXPERIMENTS

### 5.1 EXPERIMENTAL SETUPS

**Benchmark dataset.** To enable rigorous evaluation of multi-concept erasure under realistic concept relationships, we introduce ERASUREBENCH-H, a **new** Hierarchical Benchmark for Concept Erasure to reflect the hierarchical and compositional nature of concepts. While prior evaluations (Lu

et al., 2024; Fan et al., 2024; Zhao et al., 2024; Li et al., 2025) often rely on datasets such as CIFAR-10 and Imagenette, these datasets treat concepts as flat, disjoint categories and therefore cannot capture the complexity of large-scale erasure involving overlapping or nested concepts. ERASUREBENCH-H addresses this gap by organizing concepts in a *brand–series–character* hierarchy, which reflects the way unlearning requests often target groups of related concepts rather than isolated categories. This structure enables evaluation across different *concept scopes*, from broad brand-level suppression to fine-grained character-level erasure. The complete taxonomy, statistics, and curation process are detailed in Appendix A.2.1.

**Baseline and evaluation.** We benchmark against *static erasure* models, ESD (Gandikota et al., 2023), AC (Kumari et al., 2023), FMN (Zhang et al., 2024a), MACE (Lu et al., 2024), SPM (Lyu et al., 2024), and SalUn (Fan et al., 2024). Implementation details are provided in Appx. A.2.2. In our performance evaluation, we report four metrics: (i) *Erasing Effectiveness Accuracy* ($\text{Acc}_{\text{EE}}$): the rate of generated images still classified as containing the erased concept(s) by a CLIP-based classifier (lower is better); (ii) *Utility Preservation Accuracy* ($\text{Acc}_{\text{UP}}$): the rate of non-target concepts preserved in generation (higher is better); (iii) *Image Fidelity*: FID (Parmar et al., 2022) computed on all generated images against MS-COCO (Lin et al., 2015); (iv) *Harmonic Accuracy*, which combines $\text{Acc}_{\text{EE}}$ and $\text{Acc}_{\text{UP}}$ to penalize degenerate solutions, $\text{Acc}_{\text{harmonic}} = \frac{2}{\frac{1}{1-\text{Acc}_{\text{EE}}} + \frac{1}{\text{Acc}_{\text{UP}}}}$.

**Multi-concept erasure settings.** We consider two key scenarios: **(1) Erasure Scope Scaling.** In Sec. 5.2, we adopt the classic multi-concept erasure scenario used in prior work: the model is trained to erase an increasing number of concepts while each generation involves only a single target concept. This setting measures robustness as the erasure scope grows to large scale. **(2) Per-Generation Erasure Subset Scaling.** This setting evaluates performance when multiple concepts must be erased within a single generation, fixing the trained erasure scope and increasing the number of simultaneously erased concepts per generation. We realize this in two complementary ways in Sec. 5.3. (i) *Conjunctions*: we construct prompts by concatenating $N$ targets with commas and conjunctions; for example, when the per-generation erasure subset has size 3, the prompt is *"a photo of the beaver, dolphin, and otter"*. (ii) *Concept scope expansion*: leveraging ERASUREBENCH-H, we target higher-level semantic concepts (series- and brand-level) that aggregate multiple subconcepts (character-level). In this case, the per-generation erasure subset size equals the concept's scope (the number of constituent unit concepts), ranging from 1 up to 62. These scenarios directly test a model's ability to dynamically compose LoRA modules without interference.

## 5.2 ERASURE PERFORMANCE UNDER EXPANDING ERASURE SCOPE

To demonstrate the limitations of static erasure methods, we evaluate their performance as the erasure scope (*i.e.*, the total number of concepts erased) increases. We adopt CIFAR-100 as the evaluation dataset, treating each of its 100 classes as an individual concept. Models are trained to erase $\{5, 10, 15, 20\}$ concepts, but each test case involves only a single target concept per-generation (*i.e.*, erasure subset size is 1).

As a concrete example, when the per-generation erasure subset contains exactly one of the first five CIFAR-100 classes (beaver, dolphin, otter, seal, whale), we compute $\text{Acc}_{\text{UP}}$ by evaluating it on each of the latter 50 CIFAR-100 concepts and taking the mean. In this setting, DYME achieves an average Harmonic Accuracy of **90.82%** across the five single-concept erasures. Because inference is dynamic, this performance remains essentially unchanged as the total erasure scope grows while static baselines deteriorate markedly (see Fig. 1 (left)). Moreover, in detailed results, ESD and AC increasingly render generations semantically meaningless as the scope expands, which causes a large drop in $\text{Acc}_{\text{UP}}$; FMN tends to retain concepts regardless of whether they are to be erased, which causes a large drop in $\text{Acc}_{\text{EE}}$. When the erasure scope reaches 20, these methods fall to around **30%** Harmonic Accuracy, far below DYME. Detailed results are provided in Table 6.

## 5.3 ERASURE PERFORMANCE UNDER EXPANDING ERASURE SUBSET

**Increasing per-generation erasure subset by conjunctions.** Having examined scalability with respect to the erasure scope, we now turn to the complementary setting that requires composing multiple LoRA adapters per generation. As defined in Sec. 5.2, we evaluate multi-concept erasure by growing the per-generation subset size. For each $N \in \{2, 3, 4, 5\}$, we follow the benchmark's canonical class order, partition classes into contiguous 5-tuples, and for each tuple take the first $N$ classes as the target set—thus the targets for $N = 2, 3, 4, 5$ are nested (prefixes of the same ordered

| Method | 2-concept | | | 3-concept | | | 4-concept | | | 5-concept | | | FID↓ |
|---|---|---|---|---|---|---|---|---|---|---|---|---|---|
| | $\text{Acc}_{EE}$↓ | $\text{Acc}_{UP}$↑ | $\text{Acc}_{harmonic}$↑ | $\text{Acc}_{EE}$↓ | $\text{Acc}_{UP}$↑ | $\text{Acc}_{harmonic}$↑ | $\text{Acc}_{EE}$↓ | $\text{Acc}_{UP}$↑ | $\text{Acc}_{harmonic}$↑ | $\text{Acc}_{EE}$↓ | $\text{Acc}_{UP}$↑ | $\text{Acc}_{harmonic}$↑ | |
| SD (Original) | 98.50 | 70.50 | - | 99.50 | 64.00 | - | 100.00 | 37.50 | - | 100.00 | 25.50 | - | 98.54 |
| MACE | 12.00 | 27.50 | 41.90 | 17.00 | 20.50 | 32.88 | 17.50 | 15.50 | 26.10 | 14.50 | 11.50 | 20.27 | 117.26 |
| SPM | 32.50 | 60.50 | 63.81 | 36.00 | 33.00 | 43.55 | 60.00 | 21.50 | 27.97 | 53.00 | 22.00 | 29.97 | 107.90 |
| SalUn | 5.50 | 41.00 | 57.19 | 11.50 | 42.50 | 57.42 | 15.00 | 8.00 | 14.62 | 12.50 | 9.00 | 16.32 | 119.38 |
| DyME w/o ortho | 40.00 | 70.50 | 64.83 | 47.00 | 64.00 | 57.98 | 67.00 | 37.50 | 35.11 | 58.50 | 25.50 | 31.59 | 112.44 |
| DyME | **2.00** | 70.50 | **82.01** | **4.00** | 64.00 | **76.80** | **7.00** | 37.50 | **53.45** | **6.00** | 25.50 | **40.12** | 109.91 |

Table 1: Multi-concept erasure performance as per-generation erasure set increases by conjunctions on the CIFAR100 dataset.

| Method | Series (small) | | | Series (medium) | | | Series (large) | | |
|---|---|---|---|---|---|---|---|---|---|
| | $\text{Acc}_{EE}$↓ | $\text{Acc}_{UP}$↑ | $\text{Acc}_{harmonic}$↑ | $\text{Acc}_{EE}$↓ | $\text{Acc}_{UP}$↑ | $\text{Acc}_{harmonic}$↑ | $\text{Acc}_{EE}$↓ | $\text{Acc}_{UP}$↑ | $\text{Acc}_{harmonic}$↑ |
| SD (Original) | 90.50 | 72.50 | – | 93.00 | 67.00 | – | 95.50 | 51.50 | – |
| MACE | 11.50 | 19.00 | 31.28 | 7.50 | 18.00 | 30.14 | 5.00 | 7.00 | 13.04 |
| SPM | 43.00 | 61.50 | 59.16 | 52.50 | 61.00 | 53.41 | 67.50 | 44.00 | 37.39 |
| SalUn | 25.50 | 65.00 | 69.43 | 22.00 | 50.50 | 61.31 | 44.50 | 36.00 | 43.67 |
| DyME | 4.50 | 72.50 | **82.43** | 8.00 | 67.00 | **77.53** | 17.50 | 51.50 | **63.41** |

Table 2: Series-level concept erasure performance on the ERASUREBENCH-H dataset.

5-tuple). We generate 200 images per method for every target set. To keep static-erasure baselines comparable, their trained erasure scope is capped at five concepts; larger scopes cause collapse that masks method differences. Metrics follow Sec. 5.1: when calculating $\text{Acc}_{EE}$, each generated image is counted as not erased if it contains at least one of the target concepts and when computing $\text{Acc}_{UP}$, we randomly sample $N$ non-target concepts and count an image as preserved if its top-$N$ CLIP logits collectively contain all $N$ sampled non-targets.

Table 1 reports CIFAR-100 results as the per-generation erased subset grows from $N=2$ to $N=5$; Imagenette and ERASUREBENCH-H exhibit the same trend, with detailed reports in Appx. A.3.2. Across static baselines, harmonic accuracy declines monotonically with larger $N$, reflecting an increasing failure to isolate target concepts—manifested as target leakage (higher $\text{Acc}_{EE}$) or over-suppression (lower $\text{Acc}_{UP}$). In contrast, DyME maintains substantially higher harmonic accuracy for all $N$. To attribute the gains, an ablated training configuration that disables our bi-level orthogonality (DyME *(w/o ortho)* in Table 1) performs markedly worse, underscoring the role of orthogonality in achieving stable composition. We include this ablation solely to isolate the effect of orthogonality; it is *not* a proper usage of DyME. Overall, as $N$ increases, DyME yields stronger multi-target erasure with better erasure effectiveness and utility preservation. Representative qualitative cases for this setting are provided in Appx. A.3.4.

**Increasing per-generation erasure subset via concept scope expansion.** Beyond explicitly enlarging the subset via conjunction prompts (Sec. 5.3), real deployments also induce erasure subset growth implicitly when the requested concept has a broader **concept scope** (Appx. A.3.5 illustrates this growth.). Using ERASUREBENCH-H, we target higher-level concepts whose concept scope equals the number of constituent unit concepts. For each higher-level target, we generate 200 images per method and report results at the character-, series- and brand-levels: at the character level we average metrics over all characters, while at the series and brand levels we bucket targets into Small, Medium and Large by the empirical tertiles of concept scope computed separately per level. Metrics follow Sec. 5.1: $\text{Acc}_{EE}$ counts an image as *not erased* if at least one unit concept from the target appears according to the CLIP classifier; $\text{Acc}_{UP}$ matches Sec. 5.2 at the character level (size of erasure subset $= 1$) and, for series/brand levels, requires that the image's top-5 CLIP logits all fall within the corresponding non-target series or brand when classified at unit-concept granularity.

Across all three concept scope levels (character, series, and brand) DyME ranks first on harmonic accuracy (character-level: Table 9; series-level: Table 2; brand-level: Table 3), indicating the best combination of erasure effectiveness and utility preservation when the per-generation subset grows via concept scope expansion. As the scope enlarges from a single character to an entire brand, the synthesis task becomes harder (the attainable $\text{Acc}_{UP}$ ceiling drops even for the underlying generator, Stable Diffusion), so relative gaps among methods and their degradation rates are more informative than differences in raw absolute values. By these criteria, DyME consistently degrades most gracefully and maintains the strongest margins at all concept scopes, while keeping competitive fidelity.

## 5.4 ABLATION STUDY

To assess the contribution of each design component, we ablate three choices—(i) dynamic LoRA composition at inference (LoRA-C), (ii) the input-aware orthogonality constraint, and (iii) the input-

| Method | Brand (small) | | | Brand (medium) | | | Brand (large) | | |
|---|---|---|---|---|---|---|---|---|---|
| | $Acc_{EE}\downarrow$ | $Acc_{UP}\uparrow$ | $Acc_{harmonic}\uparrow$ | $Acc_{EE}\downarrow$ | $Acc_{UP}\uparrow$ | $Acc_{harmonic}\uparrow$ | $Acc_{EE}\downarrow$ | $Acc_{UP}\uparrow$ | $Acc_{harmonic}\uparrow$ |
| SD (Original) | 86.50 | 61.50 | - | 90.00 | 10.50 | - | 92.00 | 7.50 | - |
| MACE | 13.00 | 23.50 | 37.00 | 3.90 | 3.50 | 6.75 | 4.50 | 2.50 | 4.87 |
| SPM | 43.00 | 55.50 | 56.24 | 23.00 | 4.50 | 8.50 | 60.50 | 6.00 | 10.42 |
| SalUn | 37.50 | 45.50 | 52.66 | 30.50 | 5.00 | 9.33 | 49.00 | 3.50 | 6.55 |
| DyME | 6.50 | 61.50 | **74.2** | 24.50 | 10.50 | **18.44** | 51.00 | 7.50 | **13.01** |

Table 3: Brand-level concept erasure performance on the ERASUREBENCH-H dataset.

agnostic orthogonality constraint. We evaluate all configurations under the conjunction-based **erasure subset** scaling setting (Sec. 5.3) to ensure that multiple adapters must be composed per generation.

We begin with Config. 1, probing whether LoRA-C itself is essential. We replace LoRA-C with two other schemes, LoRA Merge and LoRA Switch, and report the mean across them while keeping both orthogonality terms intact. This substitution leads to a marked degradation, which is consistent with reported scalability limitations of these earlier LoRA combination techniques (Zhang et al., 2023b; Zhong et al., 2024). Next, we isolate the role of each orthogonality constraint in turn. Removing only the input-aware term (Config. 2) while retaining the input-agnostic term reveals how much ben-

| Config | LoRA-C | Orthogonality components | | | Metrics | | |
|---|---|---|---|---|---|---|---|
| | | $\mathcal{L}_{Ortho}^{In\text{-}Aware}$ | $\mathcal{L}_{Ortho}^{In\text{-}Ag}$ | PBO | $Acc_{EE}\downarrow$ | $Acc_{UP}\uparrow$ | $Acc_{harmonic}\uparrow$ |
| 1 | - | ✓ | ✓ | – | 53.25 | 70.50 | 56.22 |
| 2 | ✓ | – | ✓ | – | 34.50 | 70.50 | 67.91 |
| 3 | ✓ | ✓ | – | – | 8.50 | 70.50 | 79.64 |
| 4 | ✓ | – | – | – | 40.00 | 70.50 | 64.83 |
| 5 | ✓ | – | – | ✓ | 19.00 | 70.50 | 75.39 |
| 6 | ✓ | ✓ | – | ✓ | 8.50 | 70.50 | 79.64 |
| DYME | ✓ | ✓ | ✓ | – | **2.00** | **70.50** | **82.01** |

Table 4: **Ablation on CIFAR-100.** LoRA-C: LoRA composition. $\mathcal{L}_{Ortho}^{In\text{-}Aware}$: input-aware orthogonality constraint. $L_{Ortho}^{In\text{-}Ag}$: input-agnostic orthogonality constraint. PBO: parameter space B orthogonality constraint.

efit arises from the input-aware orthogonality constraint; symmetrically, removing only the input-agnostic term (Config. 3) exposes the contribution of it. To gauge the necessity of enforcing both simultaneously, we also consider a no-orthogonality setting (Config. 4) in which neither term is applied. Across these variants, removing or weakening either pathway reduces performance; fully removing the bi-level orthogonality yields the largest drop (see Table 1 for more details), and in this task the input-aware constraint contributes more than the input-agnostic term. Finally, we compare our representation-space constraints to a parameter-space alternative inspired by orthogonal adaptation (Po et al., 2024). Specifically, for each corresponding LoRA layer and each concept pair, we enforce zero inner product between the $B$ matrices, $B_i^\top B_j = 0$; we refer to this as the parameter-space $B$ orthogonality (PBO). We evaluate PBO as a full replacement for our bi-level orthogonality and also as a hybrid in which PBO substitutes only for the input-aware constraint. While orthogonal adaptation is helpful, it ignores inter-layer interactions within cross-attention; empirically, enforcing matrix-level $B$-orthogonality provides some gains for concept erasure but remains inferior to our bi-level orthogonality constraints. Results in Table 4 show that replacing dynamic composition hurts performance, and that bi-level orthogonality is the largest single contributor to multi-concept erasure task, with the input-aware term especially impactful.

# 6 CONCLUSION

We presented DYME, a dynamic multi-concept erasure framework for text-to-image diffusion models that reframes concept erasure as an on-demand, modular capability. By training concept-specific LoRA modules with bi-level orthogonality constraints, DYME enables composable multi-concept erasure, even as the number or granularity of targeted concepts increases. Extensive experiments on both standard (CIFAR-100, Imagenette) and newly introduced hierarchical benchmarks (ERASUREBENCH-H) demonstrate that DYME achieves significantly higher erasure effectiveness and utility preservation than existing approaches, while scaling to large, realistic erasure scenarios.

Our results highlight the importance of moving beyond static fine-tuning toward dynamic, inference-time control, and show that principled disentanglement in both feature and parameter spaces is critical for robust multi-concept erasure. We hope DYME and ERASUREBENCH-H will facilitate further progress toward practical, scalable safeguards in generative modeling.

ETHICS STATEMENT

This work studies text-to-image diffusion models by *removing* copyrighted concepts, with the goal of reducing legal and policy risk in real deployments.

**Dataset release.**   We release ERASUREBENCH-H as a *CSV taxonomy only*, containing about 300 unit-concept names (i.e., character names). It includes *no images, audio, video, bios, or identifiers*, and thus does not contain personal data or sensitive attributes. Names and groupings are used purely as string labels for research on concept erasure. We do not distribute any copyrighted media. Trademarks, if mentioned, are for referential purposes; we will honor legitimate takedown requests.

**Image generation protocol.**   To evaluate erasure quality without degenerate all-black outputs, we temporarily disabled the Stable Diffusion safety checker *during controlled offline experiments*. Prompts excluded sexually explicit, violent, or otherwise sensitive content. All generated images were used solely to compute aggregate metrics and were deleted after the experiments; we do not redistribute generated samples. Any released code/configurations will keep the safety checker *enabled by default*.

**Other ethics topics.**   This study does not involve human subjects, user data, or personally identifiable information; no IRB was required. We disclose no conflicts of interest or external sponsorship. The work does not aim to enable harmful applications; rather, it provides technical means to *restrict* the generation of copyrighted or disallowed content. We are committed to lawful, policy-compliant use of third-party models and datasets and to accurate documentation of our methods and results.

REPRODUCIBILITY STATEMENT

We have taken several steps to facilitate reproducibility. Implementation details for our method and all baselines (optimizer, learning rates, LoRA ranks/scales, training steps, sampler schedules, and composition rules) are documented in Appx. A.2.2, with evaluation metrics defined in Sec. 5.1. The full taxonomy and curation protocol for ERASUREBENCH-H are provided in Appx. A.2.1; the dataset itself is a CSV taxonomy (no images) and will be released publicly together with our cleaned codebase after submission. The code release will include configuration files that reproduce the main tables and figures, as well as the exact random seeds used to generate quantitative results and qualitative samples. Where applicable, we also provide scripts to regenerate tables/plots from saved predictions to decouple heavy compute from post-processing.

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

# A APPENDIX

## A.1 BACKGROUND

### A.1.1 LATENT DIFFUSION MODELS (LDMs)

Latent diffusion models (LDMs) perform the diffusion process in a compressed latent space rather than pixel space. Let $\mathcal{E}$ and $\mathcal{D}$ denote the encoder and decoder of a pretrained autoencoder (e.g., VAE), mapping images $x$ to latents $\mathbf{z} = \mathcal{E}(x)$ and back $\hat{x} = \mathcal{D}(\mathbf{z})$. The forward (noising) process constructs a sequence $\{\mathbf{z}_t\}_{t=0}^T$ by progressively adding Gaussian noise, while the reverse (denoising) process is learned via a conditional denoiser $\epsilon_\theta(\mathbf{z}_t, t, \mathbf{p})$, which predicts the noise at timestep $t$ given the latent $\mathbf{z}_t$ and a text prompt embedding $\mathbf{p}$. Starting from $\mathbf{z}_T \sim \mathcal{N}(\mathbf{0}, \mathbf{I})$, iterative updates using $\epsilon_\theta$ produce $\mathbf{z}_0$, which is then decoded to an image $\hat{x}_0 = \mathcal{D}(\mathbf{z}_0)$.

A common training objective is the (weighted) noise-prediction loss:

$$\mathcal{L}_{\text{diff}}(\theta) \;=\; \mathbb{E}_{x, \mathbf{p}, t, \epsilon}\Big[\big\|\epsilon - \epsilon_\theta(\mathbf{z}_t, t, \mathbf{p})\big\|_2^2\Big], \quad \text{where } \mathbf{z}_t = \alpha_t\,\mathcal{E}(x) + \sigma_t\,\epsilon, \; \epsilon \sim \mathcal{N}(\mathbf{0}, \mathbf{I}).$$

Here $\alpha_t$ and $\sigma_t$ come from the noise schedule. Conditioning on $\mathbf{p}$ enables text-guided generation; classifier-free guidance and various samplers (e.g., DDIM) are typically used at inference to trade off fidelity and diversity.

### A.1.2 CROSS-ATTENTION IN T2I MODELS

Cross-attention integrates textual context into visual latents within the U-Net backbone. Given an input hidden state $\mathbf{H} \in \mathbb{R}^{n \times d}$ (from the image pathway) and a text embedding matrix $\mathbf{T} \in \mathbb{R}^{m \times d}$, the module applies learned projections:

$$\mathbf{Q} = \mathbf{H}\,W_q, \qquad \mathbf{K} = \mathbf{T}\,W_k, \qquad \mathbf{V} = \mathbf{T}\,W_v,$$

where $W_q, W_k, W_v \in \mathbb{R}^{d \times d}$ are the query, key, and value matrices. Attention weights and outputs are computed as

$$\text{Attn}(\mathbf{H}, \mathbf{T}) = \text{softmax}\Big(\tfrac{\mathbf{Q}\mathbf{K}^\top}{\sqrt{d}}\Big)\mathbf{V}\ W_o,$$

with an output projection $W_o \in \mathbb{R}^{d \times d}$. In multi-head settings, these computations are performed head-wise and concatenated before $W_o$. Because cross-attention mixes text-derived $(W_k, W_v)$ information with image-derived queries $(W_q)$ in shared layers, it is a primary site where multiple adapters (e.g., LoRA modules) can interact—motivating our later analysis of crosstalk and orthogonality constraints.

## A.2 DATASETS AND IMPLEMENTATION DETAILS.

### A.2.1 ERASUREBENCH-H DATASET.

| Concept Scope | | |
|---|---|---|
| **Brands** | **Series** | **Characters** |
| Disney | Mickey Mouse Clubhouse | Mickey Mouse |
| | | Minnie Mouse |
| | | … |
| | Duffy and Friends | Duffy |
| | | ShellieMay |
| | | … |
| | The Lion King | Simba |
| | | Timon |
| | | … |
| DC | Justice League | Batman |
| | | Wonder Woman |
| | | … |
| | Shazam! | Shazam |
| | | … |

Table 5: Hierarchy overview of ERASUREBENCH-H (27 brands, 73 series, 300 characters)

As illustrated in Table 5 ERASUREBENCH-H organizes concepts in a brand–series–character hierarchy, supporting evaluation across semantic scopes and multi-level composition. For example, brands

(e.g., Disney, Warner Bros., DC) decompose into series (e.g., *The Lion King*, *Mickey Mouse Clubhouse*, *Looney Tunes*), which in turn decompose into characters (e.g., Simba, Mickey, Bugs Bunny, Daffy Duck). In total, the benchmark comprises 27 brands, 73 series, and 300 character-level unit concepts. We define a concept's scope as the number of unit concepts it subsumes.

The hierarchical structure serves two key purposes: (1) It enables controlled evaluation of erasure at multiple semantic levels, allowing us to test models' ability to erase high-level collective concepts (*e.g.*, "Disney character") that implicitly refer to multiple sub-concepts. (2) It supports structured analysis of semantic overlap, subset composition, and the scalability of erasure mechanisms under concept entanglement. Unlike existing flat-label datasets, ERASUREBENCH-H is specifically constructed to capture the compositional complexity of real-world content and the challenges it poses for scalable concept erasure, facilitating systematic testing under both single- and multi-concept erasure settings.

### A.2.2 IMPLEMENTATION DETAILS.

All selected baselines have official, publicly available implementations and expose interfaces that support our multiple concepts erasure setting. For completeness, Sec. 5.2 reports a comprehensive comparison across all baselines. Methods that are markedly underperforming in this setting, consistent with prior reports (Zhao et al., 2024; Zhang et al., 2024b; Li et al., 2025), are not carried forward to more complex studies. Accordingly, Sec. 5.3 focuses on the strongest static baselines (MACE (Lu et al., 2024), SPM (Lyu et al., 2024), and SalUn (Fan et al., 2024)).

All models are built on Stable Diffusion v1.4 and fine-tuned using a 50-step DDIM sampler (Song et al., 2022). Each concept-specific LoRA is trained for **20 epochs**. For the orthogonality constraints, we compute pairwise orthogonality scores across all LoRA modules and, considering computational efficiency, randomly draw 50 LoRA pairs per update. We follow the standard prompt template used in prior work, `a photo of the {target concepts}`, which makes the per-generation *erasure subset* explicit and easy to identify. Baseline methods are trained with their default configurations as reported in their respective papers. Unless otherwise stated, training uses Adam with a learning rate of $1 \times 10^{-5}$ and mini-batches of size 4. Unless otherwise noted, each per-concept adapter uses rank $r=8$ with $\alpha=r$ (effective scale $\alpha/r=1$), dropout $= 0$, and base weights are frozen (no LoRA on the text encoder), biases are not trained, and LoRA parameters are initialized from $\mathcal{N}(0, 10^{-4})$.

### A.3 ADDITIONAL RESULTS FOR MULTI-CONCEPT ERASURE

### A.3.1 ADDITIONAL RESULTS (PER CONCEPT) FOR ERASURE SCOPE SCALING ON CIFAR-100

This section reports per-class results for the scope-scaling study (Sec. 5.2). The per-generation erasure subset size is fixed to 1, and the erasure scope size varies over $\{5, 10, 15, 20\}$. As shown in Table 6, we evaluate five CIFAR-100 classes (beaver, dolphin, otter, seal, whale). For each method and scope size we report erasing effectiveness accuracy $\text{Acc}_{\text{EE}}$ (lower is better), utility preservation accuracy $\text{Acc}_{\text{UP}}$ (higher is better), and their harmonic aggregate $\text{Acc}_{\text{harmonic}}$ (higher is better). Dashes indicate results that are not available or not applicable.

### A.3.2 ADDITIONAL RESULTS FOR ERASURE SUBSET SCALING BY CONJUNCTIONS ON IMAGENETTE AND ERASUREBENCH-H

This section reports per-benchmark results corresponding to Sec. 5.3. We follow the same protocol: for each benchmark's canonical class order, we partition classes into contiguous 5-tuples and, for each subset size $N \in \{2, 3, 4, 5\}$, take the first $N$ classes as the target set (prefix nesting). For every target set and method, we generate 200 images. To keep static-erasure baselines comparable, their trained erasure scope is capped at five concepts (larger scopes collapse and obscure method differences). Metrics are computed as in Sec. 5.1: $\text{Acc}_{\text{EE}}$ counts an image as *not erased* if it contains *at least one* target concept, while $\text{Acc}_{\text{UP}}$ requires the *top-5* CLIP logits to collectively contain *all five* corresponding non-target concepts; harmonic accuracy is then the harmonic mean of the two.

Table 7 lists the Imagenette results and Table 8 lists the ERASUREBENCH-H results. Qualitatively, both benchmarks mirror the trend observed on CIFAR-100: as $N$ increases, baselines degrade, whereas DYME maintains a clear advantage. See Fig. 4 for a compact visualization of these trends.

| Method | Erasure Scope size | $\text{Acc}_{EE} \downarrow$ | | | | | $\text{Acc}_{UP} \uparrow$ | $\text{Acc}_{harmonic} \uparrow$ | | | | |
|---|---|---|---|---|---|---|---|---|---|---|---|---|
| | | beaver | dolphin | otter | seal | whale | | beaver | dolphin | otter | seal | whale |
| ESD | 5 | 7.50 | 23.00 | 21.50 | 4.50 | 11.50 | 67.13 | 77.80 | 71.73 | 72.37 | 78.84 | 76.35 |
| | 10 | 4.00 | 10.50 | 7.50 | 3.50 | 4.00 | 35.47 | 51.80 | 50.81 | 51.28 | 51.87 | 51.80 |
| | 15 | 2.50 | 1.50 | 2.00 | 2.50 | 4.00 | 10.83 | 19.49 | 19.51 | 19.50 | 19.49 | 19.46 |
| | 20 | 0.00 | 0.00 | 2.00 | 1.00 | 1.00 | 5.22 | 9.92 | 9.92 | 9.91 | 9.92 | 9.92 |
| AC | 5 | 87.50 | 89.00 | 80.00 | 88.50 | 95.50 | 88.19 | 21.90 | 19.56 | 32.61 | 20.35 | 8.56 |
| | 10 | 94.50 | 92.00 | 87.50 | 90.00 | 94.00 | 82.76 | 10.31 | 14.59 | 21.72 | 17.84 | 11.19 |
| | 15 | 92.00 | 88.50 | 83.50 | 93.50 | 92.00 | 87.11 | 14.65 | 20.32 | 27.74 | 12.10 | 14.65 |
| | 20 | 91.00 | 91.50 | 90.00 | 96.00 | 94.50 | 83.88 | 16.25 | 15.43 | 17.87 | 7.64 | 10.32 |
| FMN | 5 | 76.00 | 63.00 | 83.00 | 55.00 | 78.00 | 86.45 | 37.57 | 51.82 | 28.41 | 59.19 | 35.07 |
| | 10 | 80.50 | 67.00 | 87.00 | 59.00 | 82.00 | 79.85 | 32.00 | 46.72 | 22.36 | 54.20 | 29.38 |
| | 15 | 81.00 | 68.00 | 88.50 | 60.50 | 83.00 | 76.01 | 30.40 | 45.04 | 20.73 | 52.42 | 27.79 |
| | 20 | 79.50 | 66.00 | 86.00 | 58.00 | 81.50 | 80.00 | 33.27 | 47.72 | 23.83 | 55.08 | 30.71 |
| SPM | 5 | 11.00 | 17.00 | 15.00 | 14.00 | 16.50 | 87.30 | 88.14 | 85.10 | 86.13 | 86.65 | 85.36 |
| | 10 | 17.00 | 21.50 | 14.00 | 10.50 | 10.00 | 88.07 | 85.46 | 83.01 | 91.86 | 88.78 | 89.02 |
| | 15 | 10.50 | 25.00 | 37.50 | 16.00 | 22.00 | 84.90 | 87.14 | 79.64 | 72.00 | 84.45 | 81.30 |
| | 20 | 38.50 | 50.00 | 67.50 | 34.50 | 47.50 | 90.12 | 73.11 | 64.32 | 47.77 | 75.86 | 66.35 |
| SalUn | 5 | 8.00 | 2.50 | 5.00 | 27.00 | 20.50 | 74.14 | 82.11 | 84.23 | 83.28 | 73.57 | 76.73 |
| | 10 | 3.00 | 4.00 | 0.00 | 6.00 | 17.00 | 40.68 | 57.32 | 57.14 | 57.83 | 56.79 | 54.60 |
| | 15 | 1.50 | 6.00 | 0.50 | 4.00 | 12.00 | 17.61 | 29.88 | 29.66 | 29.92 | 29.76 | 29.35 |
| | 20 | 0.00 | 3.00 | 3.50 | 5.50 | 4.00 | 11.52 | 20.66 | 20.59 | 20.58 | 20.54 | 20.57 |
| MACE | 5 | 1.00 | 12.00 | 0.00 | 5.00 | 22.00 | 78.29 | 87.44 | 82.86 | 87.82 | 85.84 | 78.14 |
| | 10 | 1.00 | 14.00 | 4.50 | 7.50 | 17.00 | 46.63 | 63.40 | 60.47 | 62.66 | 62.00 | 59.71 |
| | 15 | 1.50 | 16.00 | 8.50 | 4.50 | 12.00 | 38.20 | 55.05 | 52.52 | 53.90 | 54.57 | 53.27 |
| | 20 | 1.00 | 5.50 | 4.00 | 3.00 | 2.00 | 14.66 | 25.54 | 25.19 | 25.44 | 25.47 | 25.50 |
| DYME | 5 10 15 20 | 1.00 | 13.00 | 0.50 | 6.00 | 22.00 | **90.52** | **94.57** | **88.72** | **94.80** | **92.22** | **83.79** |
| SD | 0 | 96.00 | 97.50 | 94.50 | 98.00 | 98.00 | 90.52 | – | – | – | – | – |

Table 6: **Erasure Scope scaling on CIFAR-100 (per-class view).** Per-generation erasure subset size is 1. "Erasure Scope size" denotes the number of concepts the model is trained to erase (the erasure scope). Columns list per-class $\text{Acc}_{EE}$ (lower is better), overall $\text{Acc}_{UP}$ (higher is better), and per-class $\text{Acc}_{harmonic}$ (higher is better). Evaluated on five CIFAR-100 classes: beaver, dolphin, otter, seal, whale. "SD" is the unmodified Stable Diffusion baseline (no erasure). Dashes indicate results not available or not applicable.

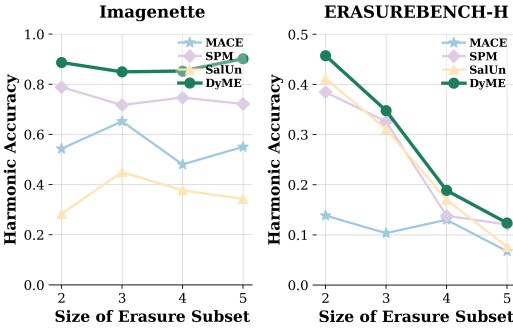

Figure 4: Erasure subset scaling by conjunction prompts on Imagenette and ERASUREBENCH-H.

### A.3.3 ADDITIONAL RESULTS FOR ERASURE SUBSET SCALING VIA CONCEPT SCOPE EXPANSION

This subsection complements Sec. 5.3 by reporting the *character-level* case, where the concept scope equals 1 and thus the per-generation *erasure subset* size is fixed at $N=1$. For each character and method, we generate 200 images and compute metrics as in Sec. 5.1: $\text{Acc}_{EE}$ (erasure effectiveness), $\text{Acc}_{UP}$ (utility preservation), and their harmonic mean. Table 9 summarizes character-level results. As expected for $N=1$, absolute performance is higher than in higher concept scope settings; nev-

| Method | 2-concept | | | 3-concept | | | 4-concept | | | 5-concept | | |
|---|---|---|---|---|---|---|---|---|---|---|---|---|
| | $Acc_{EE}\downarrow$ | $Acc_{UP}\uparrow$ | $Acc_{harmonic}\uparrow$ | $Acc_{EE}\downarrow$ | $Acc_{UP}\uparrow$ | $Acc_{harmonic}\uparrow$ | $Acc_{EE}\downarrow$ | $Acc_{UP}\uparrow$ | $Acc_{harmonic}\uparrow$ | $Acc_{EE}\downarrow$ | $Acc_{UP}\uparrow$ | $Acc_{harmonic}\uparrow$ |
| SD (Original) | 92.50 | 82.00 | - | 97.00 | 76.50 | - | 94.00 | 77.00 | - | 90.00 | 83.50 | - |
| MACE | 8.00 | 38.50 | 54.28 | 4.50 | 49.50 | 65.20 | 8.00 | 32.50 | 48.03 | 9.50 | 39.50 | 55.00 |
| SPM | 10.50 | 70.50 | 78.87 | 18.50 | 64.00 | 71.70 | 14.00 | 66.00 | 74.68 | 16.50 | 63.50 | 72.14 |
| SalUn | 26.00 | 17.50 | 28.31 | 21.50 | 31.50 | 44.96 | 24.50 | 24.50 | 37.67 | 22.00 | 22.00 | 34.32 |
| DyME w/o ortho | 76.50 | 82.00 | 36.53 | 62.50 | 76.50 | 50.33 | 70.50 | 77.00 | 42.66 | 53.00 | 83.50 | 60.15 |
| DyME | **3.50** | **82.00** | **88.66** | **4.50** | **76.50** | **84.95** | **4.50** | **77.00** | **85.26** | **2.00** | **83.50** | **90.17** |

Table 7: Multi-concept erasure performance as per-generation erasure set increases by conjunctions on the Imagenette dataset.

| Method | 2-concept | | | 3-concept | | | 4-concept | | | 5-concept | | |
|---|---|---|---|---|---|---|---|---|---|---|---|---|
| | $Acc_{EE}\downarrow$ | $Acc_{UP}\uparrow$ | $Acc_{harmonic}\uparrow$ | $Acc_{EE}\downarrow$ | $Acc_{UP}\uparrow$ | $Acc_{harmonic}\uparrow$ | $Acc_{EE}\downarrow$ | $Acc_{UP}\uparrow$ | $Acc_{harmonic}\uparrow$ | $Acc_{EE}\downarrow$ | $Acc_{UP}\uparrow$ | $Acc_{harmonic}\uparrow$ |
| SD (Original) | 62.50 | 30.50 | - | 74.0 | 21.50 | - | 81.00 | 10.50 | - | 84.50 | 7.00 | - |
| MACE | 9.50 | 7.50 | 13.85 | 14.50 | 5.50 | 10.34 | 8.00 | 7.00 | 13.01 | 9.50 | 3.50 | 6.74 |
| SPM | 10.50 | 24.50 | 38.47 | 20.50 | 20.50 | 32.59 | 14.00 | 7.50 | 13.80 | 16.50 | 6.50 | 12.06 |
| SalUn | 13.50 | 27.00 | 41.15 | 14.00 | 19.00 | 31.12 | 18.50 | 9.50 | 17.02 | 22.00 | 4.00 | 7.61 |
| DyME w/o ortho | 56.50 | 30.50 | 35.86 | 52.50 | 21.50 | 29.60 | 60.00 | 10.50 | 16.63 | 47.00 | 7.00 | 12.37 |
| DyME | **8.50** | **30.50** | **45.75** | **9.00** | **21.50** | **34.78** | **7.00** | **10.50** | **18.87** | **7.50** | **7.00** | **13.02** |

Table 8: Multi-concept erasure performance as per-generation erasure set increases by conjunctions on the ERASUREBENCH-H dataset.

| Method | Character-level | | | FID↓ |
|---|---|---|---|---|
| | $Acc_{EE}\downarrow$ | $Acc_{UP}\uparrow$ | $Acc_{harmonic}\uparrow$ | |
| SD (Original) | 72.50 | 71.20 | - | 117.01 |
| MACE | 7.50 | 34.40 | 50.15 | 140.19 |
| SPM | 27.00 | 61.60 | 66.82 | 134.57 |
| SalUn | 8.50 | 21.00 | 34.16 | 134.57 |
| DyME | **7.50** | **71.20** | **80.46** | **133.04** |

Table 9: Character-level concept erasure performance on the ERASUREBENCH-H dataset.

ertheless, DyME maintains the best trade-off between erasure effectiveness and utility, consistent with the trends in the main text.

### A.3.4 QUALITATIVE COMPARISON FOR ERASED SUBSET SCALING BY CONJUNCTIONS

This subsection complements Sec. 5.3 with qualitative examples under the same setting. Baselines are trained with an *erasure scope* of 5 and evaluated on conjunction prompts with a per-generation *erasure subset* size of $N = 2$.

As shown in Fig. 5, panel (a) illustrates *erasure effectiveness*: all targets in the subset should be suppressed; any visible target indicates leakage. Panel (b) illustrates *utility preservation*: the specified non-target concepts must be preserved simultaneously. Rows correspond to the same prompt and random seed; columns compare DyME with static-erasure baselines.

Across prompts, static baselines either exhibit target leakage or over-suppress non-targets. In contrast, DyME reliably removes all concepts in the subset and, by virtue of its dynamic erasure, refrains from activating any LoRA for non-target concepts, thereby matching the base Stable Diffusion output for those elements.

### A.3.5 CASE STUDY: CONCEPT-SCOPE EXPANSION AND PER-GENERATION ERASURE SUBSET

We illustrate how *concept-scope expansion* enlarges the per-generation *erasure subset* using a brand–series–character hierachy (e.g., the brand is Disney; the series is Mickey Mouse Clubhouse; the character is Mickey Mouse). At the character level, the subset size is 1; at the series level it equals the number of characters in the series; at the brand level it equals the number of unit concepts under the brand. For each level we generate images with the prompts shown in Fig. 6 and apply DyME.

As shown in Fig. 6, enlarging the concept scope from character to series and brand leads to leakage of an increasing number of unit concepts within the higher-level category. Consequently, the per-generation *erasure subset* expands, and DyME must dynamically activate and compose more LoRA adapters to suppress all implicated units. These observations validate the protocol of scaling the *erasure subset* via *concept-scope* expansion and underscore the necessity of the hierarchical benchmark ERASUREBENCH-H that makes concept scope explicit.

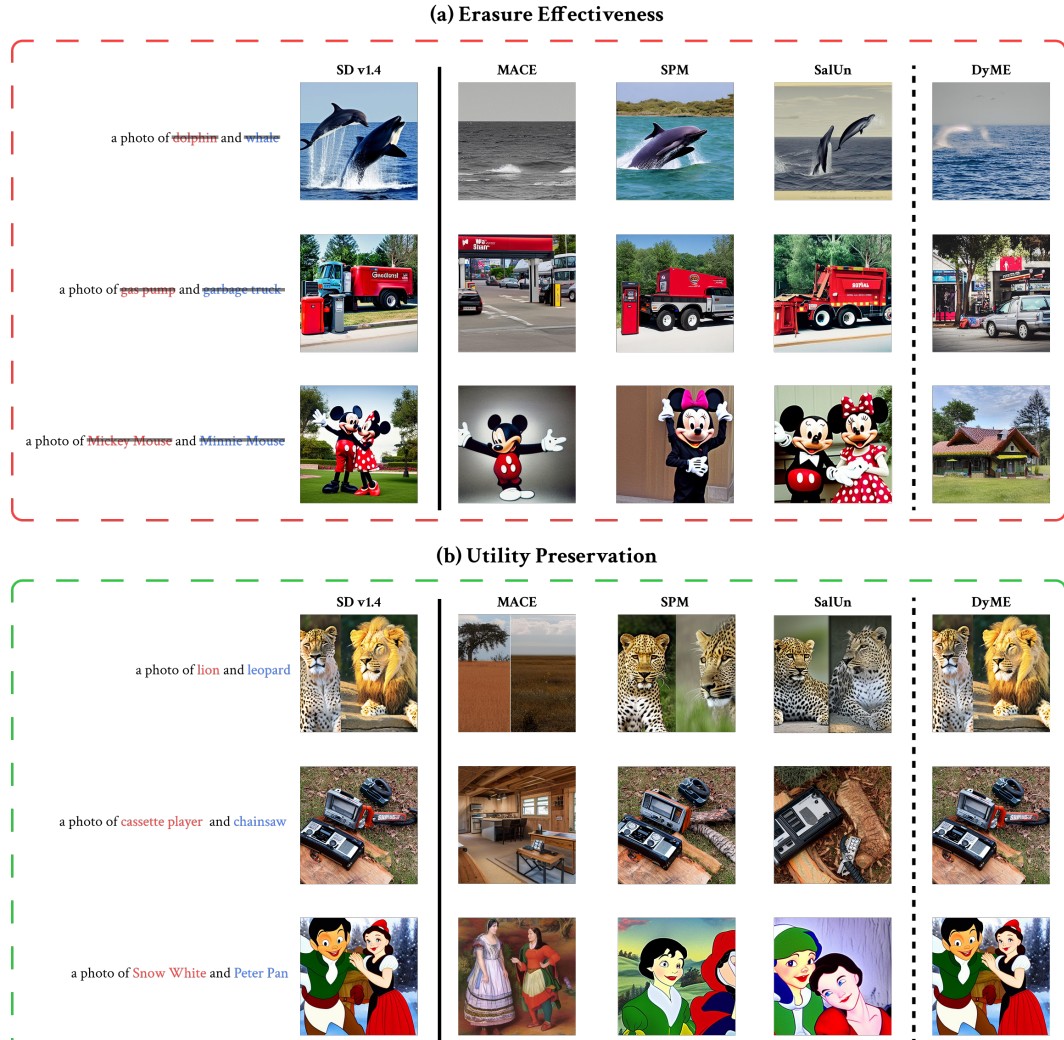

Figure 5: Qualitative comparison: the size of erased subset scaling to 2 by conjunctions. (a) *Erasure effectiveness*: target concepts should be removed; any residual target indicates leakage. Across prompts, static baselines either exhibit target leakage or over-suppress non-targets. (b) *Utility preservation*: all specified non-target concepts should appear simultaneously. DYME, by virtue of its dynamic erasure, refrains from activating any LoRA for non-target concepts, thereby matching the base Stable Diffusion output for those elements. DYME is compared against baselines and the images on the same row are generated using the same random seed.

### A.4 THE USE OF LARGE LANGUAGE MODELS

We used a large language model (e.g., ChatGPT) only for copy-editing: checking spelling, grammar, punctuation, and minor stylistic issues. No substantive content (ideas, claims, equations, methods, analyses, results, figures, tables, code, or data) was generated or modified by an LLM. All edits were reviewed and accepted by the authors, who take full responsibility for the contents of this manuscript. LLMs are not eligible for authorship.

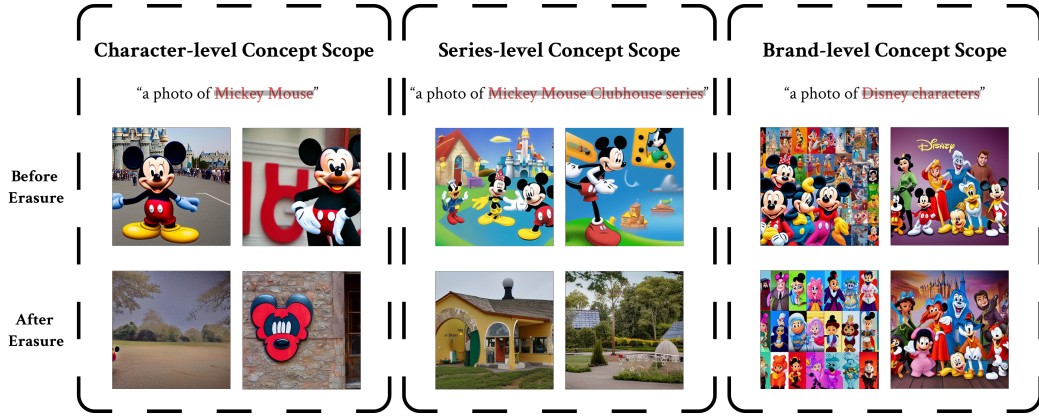

Figure 6: **Concept-scope expansion increases the per-generation erasure subset.** Left to right: character-, series-, and brand-level concept scopes (prompts shown within each level). Top: generations before erasure; bottom: after applying DYME. As concept scope grows, the number of unit concepts to suppress per generation (the erasure subset) increases.

