# OpenReview forum: "DyME: Dynamic Multi-Concept Erasure in Diffusion Models with Bi-Level Orthogonal LoRA Adaptation"
_ICLR.cc/2026/Conference — ICLR 2026 Conference Withdrawn Submission_

### Official Review · Reviewer_SrNi · 2025-10-21

**Soundness:** 3
**Presentation:** 2
**Contribution:** 2
**Rating:** 4
**Confidence:** 3

**Summary:**

This paper proposes a dynamic concept erasure framework for multi-concept unlearning. The proposed framework first learns concept-specific LoRA for each concept. Then the framework learns to dynamically compose the LoRAs corresponding to the target concepts during inference. The experiments are conducted on various commonly used benchmarks. In addition, this paper also proposes an ErasureBench-H benchmark for more comprehensive multi-concept evaluation.

**Strengths:**

1. Multi-concept erasure is crucial for real-world trustworthy generative AI deployments.
2. The proposed method is reasonable to handle the dynamic scenarios during inference.
3. The overall paper is easy to follow.

**Weaknesses:**

1. The proposed framework applies LoRA adapters to encode the specific concepts. However, in the real-world multi-concept erasure setting, the number of concepts would be large. It would cause the memory issues that the users cannot store such a large amount of adapters.
2. Step 3 of the proposed method requires the framework to first train all concept LoRAs jointly. However, in real-world scenarios, it is possible that the user wants to add novel concepts beyond the training set. It will require re-training once novel concepts appear, which might not be practical.
3. While this paper enforces each concept to be non-overlapping, it remains hard to handle the rephrased prompts that would cause robustness issues, allowing attackers to easily recover the concept by paraphrasing the target prompts.

**Questions:**

1. This paper contains no qualitative visualization in the main paper. It would be beneficial to demonstrate the visualization for concept erasure on a large number of multiple concepts (e.g., 5 or more concepts).

---

> ### Author Response · Authors · 2025-11-25
> **Response to Reviewer #SrNi**
>
> We thank the reviewer for recognizing the importance of DyME for trustworthy generative AI deployments, the reasonableness of our dynamic composition framework with concept-specific LoRAs, and the clarity of the overall presentation and experimental evaluation. We address your questions and suggestions one by one below.
>
> > **W1.** Scalability of storing a large number of LoRA adapters
>
> **A1.** We appreciate the reviewer’s concern regarding scalability. In practice, the memory footprint of our LoRA-based design is very small. Each concept-specific adapter is approximately **6.2 MB**, which is several orders of magnitude smaller than storing full model replicas. This makes the approach highly scalable even when the number of concepts grows to hundreds or thousands. We will make this explicit by adding a detailed breakdown of memory cost in the revision.
>
> > **W2.** Practicality when users introduce novel concepts
>
> **A2.** We thank the reviewer for highlighting this real-world scenario. Supporting incremental addition of new concepts is a central motivation of our design. Traditional static erasure methods require full model retraining whenever the erased set expands, which is expensive and often infeasible. Our dynamic scheme only trains lightweight LoRA adapters for the new concepts. This significantly reduces the computational cost. In our experiment on 20 new concepts, the training time required by our approach is **8.64 hours**. This is substantially lower than prior methods such as SPM which requires **24.6 hours** and ESD which requires **14 hours**.
>
> > **W3.** Robustness to rephrased prompts
>
> **A3.** We acknowledge that adversarial robustness (e.g., "jailbreaking" via paraphrasing) is a critical open challenge in this field. However, we respectfully submit that this is orthogonal to the contribution of our paper, which focuses on scalability and interference. Our framework operates under the standard "cooperative usage" assumption common in concept erasure works (e.g., ESD, UCE). This setting targets service providers (such as Midjourney or OpenAI) that wish to comply with regulations by removing protected concepts from the model's base distribution.
>
> In adversarial scenarios, a determined attacker can bypass any current unlearning method (or simply use an original unprotected model). While we agree that robustness is vital, our contribution is solving the "interference" problem that prevents providers from erasing more than a handful of concepts at a time. We will explicitly clarify this scope and limitation in the final revision.
>
> > **Q1.** Lack of qualitative visualization for large numbers of concepts
>
> We appreciate the reviewer’s request for qualitative demonstrations. We want to clarify that the visualizations for large-scale erasure are provided in the appendix.
>
> - **Figure 5** ([link](https://anonymous.4open.science/r/ForAnonymousGithub-C9D0/Exp2ImageComparison.png)):
>   Presents qualitative comparisons for the large-scale setting where the erasure scope size is 20 concepts, and the erased subset size is two, using conjunctive prompts.
>
> - **Figure 6** ([link](https://anonymous.4open.science/r/ForAnonymousGithub-C9D0/Exp3Case.png)):
>   Further illustrates the erasure quality across different semantic levels, ranging from character-level to series-level and brand-level concepts. E.g., In the figure “Mickey Clubhouse sries” or “Disney characters” implicitly involve up to 15 and 62 unit character concepts respectively.
>
> These examples demonstrate that our method consistently removes the targeted concepts even when the erased set is large and heterogeneous, while maintaining image quality and preserving unrelated attributes. We will highlight these results more clearly in the main text to better highlight the method's qualitative performance on large-scale tasks.

---

### Official Review · Reviewer_QDp2 · 2025-10-22

**Soundness:** 1
**Presentation:** 2
**Contribution:** 2
**Rating:** 2
**Confidence:** 5

**Summary:**

This paper proposes DYME, a dynamic multi-concept erasure framework for text-to-image diffusion models that allows on-demand suppression of copyrighted or sensitive concepts. Instead of static fine-tuning, DYME trains lightweight, concept-specific LoRA adapters and dynamically composes only those needed at inference. To mitigate interference among multiple erased concepts, it introduces bi-level orthogonality constraints at both the feature and parameter levels. The authors also present ERASUREBENCH-H, a hierarchical benchmark reflecting real-world brand–series–character relationships. Extensive experiments on CIFAR-100, Imagenette, and ERASUREBENCH-H show that DYME achieves superior erasure fidelity and utility preservation compared to prior static approaches, effectively scaling to large and overlapping concept sets.

**Strengths:**

1. The paper introduces a new concept erasure setting, where multiple target concepts can be erased simultaneously during inference, reflecting more realistic real-world scenarios.
2. It proposes bi-level orthogonality constraints, which effectively enable the stable composition of multiple concept-specific LoRA modules at inference without interference.
3. The paper presents ERASUREBENCH-H, a hierarchical and semantically structured benchmark that enables comprehensive and realistic evaluation of multi-concept erasure performance

**Weaknesses:**

1. In Figure 1, the performance of MACE appears inconsistent with its original paper, where it successfully erased 100 concepts (e.g., 100 celebrities) while maintaining high harmonic accuracy. Here, only up to 20 concepts are erased, and the results for MACE seem unexpectedly poor. This might be due to differences in the CIFAR-100 setting, where protected concepts may not have been properly included in the retention set. The authors are encouraged to revisit their reproduction settings. In addition, SPM should be categorized under Dynamic Erasure (DE), as described in Section 3.3 of its paper.
2. In line 73, the claim that static approaches “reduce diversity and degrade fidelity” lacks supporting evidence. In contrast, MACE demonstrates good fidelity preservation even under multi-concept erasure, which contradicts this statement.
3. In the Introduction, the paper does not clearly define the difference between static erasure paradigm and dynamic erasure paradigm, even though these are the core concepts that underpin the paper’s motivation. This omission makes it difficult for readers to fully grasp the main idea.
4. The major concern lies in the motivation of the proposed task setup. The authors emphasize that the weakness of static erasure lies in its inability to dynamically select concepts during inference, while all concepts are erased during training. However, the fundamental goal of concept erasure is to permanently remove target semantics from the model parameters without affecting others. The proposed dynamic erasure is unrealistic in white-box settings (e.g., Stable Diffusion), since LoRA adapters are external and not merged into model weights — an attacker could easily bypass DYME by modifying the inference code directly.
5. Although the proposed method might have value in black-box deployment, similar behavior could be achieved with simpler mechanisms. Since DYME selects LoRA modules based on keyword matching, a trivial baseline could simply detect forbidden keywords in prompts and return a blank or noisy image otherwise generating normally, achieving equivalent or even better results (i.e., AccEE = 0 and unchanged AccUP).
6. The proposed Dynamic Composition at Inference (Sec. 4.2) relies on explicit keyword matching, which fails to address implicit concepts, such as those in the I2P benchmark [1]. Handling NSFW or implicit concepts is a critical aspect of practical concept erasure that this method does not consider.
7. In line 106, the paper claims “this is the first work to systematically investigate multi-concept erasure scalability in diffusion models,” but prior studies such as [2, 3] have already discussed similar topics.
8. In line 241, the symbol j is undefined.
9. In Equation (3), the loss formulation seems to encourage opposite directions (cosine similarity = –1) rather than orthogonality (cosine similarity = 0), which contradicts the intended behavior.
10. The experimental comparisons include only SPM and MACE (both CVPR 2024). More recent and relevant baselines with similar motivations are missing and should be added for completeness.
11. The experiments in Section 5.2 only cover up to 20 erased concepts, while MACE has demonstrated the ability to erase 100 concepts. Although the dynamic approach may scale better, expanding the number of erased conceptswould strengthen the motivation and claims.
12. The ERASUREBENCH-H benchmark lacks sufficient detail and description in the main text; it is hard to understand its structure and usage without referring to the appendix.
13. In Table 4 (Config 1), it is unclear why AccUP remains 70.50 after removing LoRA-C. This inconsistency needs clarification or additional explanation.



[1] Safe Latent Diffusion: Mitigating Inappropriate Degeneration in Diffusion Models, CVPR23

[2] Localized Concept Erasure for Text-to-Image Diffusion Models Using Training-Free Gated Low-Rank Adaptation, CVPR25

[3] Erasing Concept Combination from Text-to-Image Diffusion Model, ICLR25

**Questions:**

See the weakness aprt.

---

> ### Author Response · Authors · 2025-11-25
> **Response to Reviewer #QDp2 (Part 1 of 3)**
>
> We thank the reviewer for emphasizing the realism of our new multi-concept erasure setting, the effectiveness of DyME, and the value of the ERASUREBENCH-H benchmark. We respond to each concern point-by-point below. Since the review provides many constructive suggestions that we would like to address, our response is divided into three parts due to character limits. This comment is the first part.
>
> > **W1.**: Inconsistent performance reports of MACE and category of SPM.
>
> **A1.** To clarify the inconsistency between MACE’s reported high accuracy on 100 concepts and our report of MACE’s low accuracy on 20 concepts: Their report is based on celebrity erasure, not on CIFAR-100 object erasure or character erasure using our dataset. We agree with your hypothesis that this discrepancy might be due to differences in the CIFAR-100 dataset, implying that different types of concepts (e.g., faces vs. objects) may affect the model's effectiveness. To verify this, we conducted a new experiment on celebrity erasure. The results show that when it comes to celebrity erasure, even methods like ESD and FMN—which are widely reported in related work to fail easily in multi-concept erasure—could still maintain a harmonic accuracy of 22.72% and 39.92% respectively, after erasing 50 celebrities. In contrast, they reached less than 20% accuracy after erasing just 10-15 CIFAR-100 objects. Thus, we believe that the capability regarding the number of erasable concepts should clearly specify which dataset corresponds to the erasure task. We also confirm that we strictly followed MACE’s official GitHub instructions.
>
> Regarding SPM’s categorization, we agree SPM possesses a dynamic erasure mechanism named Facilitated Transport (FT). FT computes the transmission rate for each SPM for every prompt according to keyword matching and CLIP score matching. In Sec. 5.2 we used SPM’s official implementation where their dynamic erasure mechanism is disabled (maybe due to high inference latency), so it acts as a static erasure method, and we reported this implementation consistently in Section 3.3. Now we also added SPM’s performance under dynamic erasure settings. Still, dynamic SPM’s performance (harmonic accuracy) drops from 86.28% to 40.02% when the erasure scope size scales from 1 to 50. The main reason could be that each concept-specific SPM’s transmission rate (how heavily it is loaded onto the DM) is the maximum of keyword match and CLIP score; even when a prompt does not contain the keyword of that concept, the CLIP score rarely becomes 0, so all trained SPMs are still loaded.
>
> Regarding the reproduction settings of MACE, we confirm that we strictly followed MACE’s official GitHub instructions. The detail is that MACE uses two parameters in the training configuration file, prior_preservation_cache and preserve_prompt, to control which concepts are treated as protected and therefore included in the retention set. We loaded the model cache provided by MACE and placed the unerased concept set in this file accordingly.
>
> > **W2.**: Claim of “fidelity” is inconsistent.
>
> **A2.** We want to clarify we used the term “fidelity” to refer to the preservation of unerased concepts, which means the ability of the generated images to normally present unerased concepts. Fidelity in this claim is not referring to the FID score (the realistic texture of generated images). MACE would erase untargeted concepts (as shown in Sec. 5.2 the generation accuracy for unerased concepts drops to 14.66% after erasing 20 object concepts). We will clarify this in the paper.
>
> > **W3.**: Definition of static erasure and dynamic erasure
>
> **A3.** We appreciate the reviewer’s comment and will clarify both definitions in the Introduction.
>
> - Static erasure paradigm. A static erasure method fine-tunes a model once on a fixed target set. The resulting checkpoint always erases this entire set during inference and cannot vary the erased subset on a per-prompt basis. In other words, suppression is all-or-nothing: if the checkpoint erases concepts A, B, and C, every generation will suppress A, B, and C regardless of what the user actually wants to remove.
>
> - Dynamic erasure paradigm. A dynamic erasure method trains a collection of modular erasure components but activates only the components relevant to each prompt at inference time. This allows the system to erase concept A for one request, concepts A and C for another, and none for a third, all using the same underlying model. By suppressing only the required concepts for each request, dynamic erasure minimizes collateral damage and better preserves generation quality for untargeted concepts.

---

> ### Author Response · Authors · 2025-11-25
> **Response to Reviewer #QDp2 (Part 2 of 3)**
>
> We thank the reviewer again. Our response is divided into three parts due to character limits. This comment is the second part.
>
> > **W4.**: Adversarial white box settings
>
> **A4.** We thank the reviewer for raising this point. Our goal is not to permanently remove concepts from the core model parameters but to support flexible and policy-driven erasure that adapts to continuously evolving requirements. Real-world deployments often face diverse and changing content rules. Examples include different definitions of nudity across countries, religion-specific prohibitions, and region-dependent political restrictions. Under such diversity, static permanent erasure would require maintaining a large and ever-growing set of model variants, which is impractical.
>
> DyME targets a realistic deployment scenario where a cloud provider offers controlled generation services and can enforce adapter activation at inference time. In this environment, dynamic, selective erasure provides substantially better policy alignment and significantly reduces collateral degradation of benign concepts.
>
> We agree that DyME does not prevent an adversary who can run the model fully offline and bypass adapter activation. This limitation is inherent to all modular or external component based defenses. We will clarify in the revision that DyME is intended for controlled cloud based deployments rather than adversarial white box settings.
>
> > **W5.**: Potential trivial baseline
>
> **A5.** We agree that this trivial baseline could reach a very good AccEE = 0. However, we also need to consider the realistic texture of the generated images.
>
> In real-world products, users require a semantically realistic image rather than a corrupted output such as blank/noisy images because of their downstream workflows—such as UI design tools, educational content generation, entertainment applications, and assistive technologies—that expect coherent visual outputs. This is why existing erasure techniques strive to modify only harmful content while preserving benign concept and visual texture, rather than collapsing the output to blank or noise.
>
> > **W6.**: Relying on explicit keyword matching
>
> **A6.**: Thank you for the concerns about DyME’s inference mechanisms. We want to clarify that DyME does not use explicit keyword matching to decide which concept-specific LoRAs should be activated at inference. We assume that DyME users know what the erasure subset is and which LoRAs to activate. In line with our answer A4, this means that DyME does not target the scenario of preventing an adversary who can run the model fully offline or conduct advanced prompt engineering (e.g., using prompts in I2P) to bypass concept erasure. DyME instead targets cloud providers that offer controlled services and assume cooperative usage. This means that for implicit concepts, DyME users are still clear about which LoRA should be used; for example, for the I2P benchmark you mentioned, users know that we need to activate the LoRA that erases “nudity.”
>
> > **W9.**: Opposite objective function
>
> **A9.** Thanks for the comment, we will clarify that our loss function design in Equation (3) is to minimize the minus-orthogonality term, which is equivalent to maximizing the orthogonality score during training. We encourage orthogonality to be large to reduce the crosstalk between LoRA modules.
>
> > **W10.**: Missing comparison to contemporaneous SOTA methods.
>
> **A10.** Thanks for suggesting a comparison with SOTA approaches. In the revision, we have compared our proposed method with two recent SOTA methods: CoGFD (ICLR 2025) and GLoCE (CVPR 2025). CoGFD performs concept combination erasure using an LLM-induced logical graph, and GLoCE performs training-free localized erasure based on SVD-derived low-rank updates.
>
> We evaluated both methods under the erasure scope scaling task in Section 5.2. When the erasure scope is 50 concepts, the harmonic accuracy of CoGFD and GLoCE drops to 25.92 percent and 36.04 percent, while our proposed DyME achieves 90.82 percent under the same setting.
>
> The performance gap arises because neither CoGFD nor GLoCE is designed for large-scale multi-concept erasure. CoGFD focuses on a small set of high-level combinations, and its logic graph does not scale to dozens or hundreds of unit concepts whose combinations grow combinatorially. GLoCE is effective for single or a few concept erasure, but it composes multiple gated low-rank modules without mechanisms to prevent cross-concept interference. When multiple gates are activated simultaneously, their updates accumulate in shared feature spaces, degrading performance. DyME directly addresses these scalability issues, which explains its advantage in large erasure scopes.

---

> ### Author Response · Authors · 2025-11-25
> **Response to Reviewer #QDp2 (Part 3 of 3)**
>
> We thank the reviewer again. Our response is divided into three parts due to character limits. This comment is the third part.
>
> > **W11.**: Extend to larger ersure scope.
>
> **A11.** We appreciate the reviewer’s suggestion. We extend the erasure scope from 20 to 50 concepts on CIFAR-100, as suggested by the reviewer. At this size, all baselines collapse sharply even when the erasure subset size is only 1, which means each per-generation prompt involves only one erased concept. For this evaluation, we treat the first five CIFAR-100 concepts from erasure scope, construct prompts that include each concept in turn, generate images, and compute accuracy for each class; we then report the average harmonic accuracy (Acc_harmonic) in the table below.
>
> | Method | Average Acc_harmonic ↑ |
> |--------|------------------------|
> | ESD    | 6.11%                  |
> | AC     | 16.56%                 |
> | FMN    | 24.58%                 |
> | SPM    | 40.02%                 |
> | SalUn  | 16.78%                 |
> | MACE   | 19.73%                 |
> | GLoCE  | 36.04%                 |
> | CoGFD  | 25.93%                 |
> | **DyME**   | **90.82%**                 |
>
> > **W12.**: New benchmark detals
>
> **A12.** We want to clarify the details of ERASUREBENCH-H. ERASUREBENCH-H organizes concepts in a brand–series–character hierarchy, supporting evaluation across semantic scopes and multi-level composition. For example, brands (e.g., Disney, Warner Bros., DC) are decomposed into series (e.g., The Lion King, Mickey Mouse Clubhouse, Looney Tunes), which in turn are decomposed into characters (e.g., Simba, Mickey, Bugs Bunny, Daffy Duck). In total, the benchmark comprises 27 brands, 73 series, and 300 character-level unit concepts. This facilitates the deployment of experiments scaling the erasure subset based on the semantic scope of the concepts.
>
> > **W13.**: Unchanged $Acc_{UP}$ in ablations
>
> **A13.** We want to clarify that AccUP remains 70.50 after removing LoRA-C in the ablation study, because here AccUP is obtained from our model with the LoRA deactivated when generating unerased concepts, so we have the same generation accuracy as the original SD v1.4. Here, removing LoRA-C does not mean we no longer use any LoRA adapters, but that we load multiple LoRAs using another method named LoRA-Merge instead of LoRA-C. Since we are still using LoRAs, we can still deactivate them. Therefore, AccUP remains 70.50, consistent with SD v1.4.

---

> > ### Comment · Reviewer_QDp2 · 2025-11-27
> >
> > Thank you for the detailed response. I appreciate the clarifications, but after reading W6 and the related discussion, several core concerns remain unresolved.
> >
> > First, regarding the application scenario described in W6, I still find this setting insufficiently practical. If the goal is simply to prevent the model from generating a specific concept, a user or a cloud provider in a controlled setting could directly modify or sanitize the prompt. It is unclear why loading LoRA adapters at inference time would be preferable to trivial alternatives such as keyword removal or prompt rewriting. The proposed use case seems to introduce unnecessary complexity without offering additional guarantees.
> >
> > Second, I believe the paper fundamentally misformulates the nature of concept erasure. Concept erasure is inherently a safety problem, not a user-choice or on-demand control problem. In practice, whether a concept should or should not be generated is typically determined by the model provider or by regulatory requirements, rather than by the end user. Under this framing, the ability for users to dynamically choose which concepts to erase is not only unnecessary but also potentially counterproductive. More importantly, in open-source or white-box settings such as Stable Diffusion checkpoints, external LoRA adapters provide no actual safety guarantees because any adversarial user can simply bypass the adapter loading mechanism. This makes the proposed dynamic paradigm unsuitable for the scenarios where concept erasure is most needed.
> >
> > The authors emphasize in W4 that DyME is intended for cloud-provider–controlled deployments. However, in such closed and provider-controlled environments, much simpler and more reliable alternatives exist. As mentioned in my earlier comments in W5, a provider can block or rewrite forbidden keywords at the prompt level, or return a blank or noisy image when disallowed content is detected. Both strategies trivially achieve AccEE = 0 and maintain AccUP, and they do so without incurring the overhead or complexity of dynamic LoRA composition. The authors argue that generating a semantically realistic image is more desirable, yet when the user explicitly requests NSFW or otherwise prohibited content, refusing to generate a realistic image is precisely the intended and acceptable behavior in safety enforcement.
> >
> > For these reasons, I believe the paper still does not convincingly justify why dynamic on-demand erasure via modular LoRA composition is necessary, why it is superior to simpler control mechanisms, or how it aligns with realistic deployment and threat models. Based on these inherent limitations, I will maintain my current rating.

---

> > > ### Author Response · Authors · 2025-11-29
> > > **Follow-Up Response to Reviewer QDp2**
> > >
> > > We thank the reviewer for the careful follow-up and for the thoughtful suggestions regarding deployment and safety considerations. Below, we respond to the new comments point by point.
> > >
> > > > **Q1.**  Why DyME would be preferable to trivial alternatives such as prompt rewriting?
> > >
> > > **A1.** Thanks for this comment. We fully acknowledge that prompt-level solutions (keyword filtering, rewriting) are simpler to deploy. However, we view these mechanisms as orthogonal rather than as mutually exclusive substitutes to parameter-level concept erasure such as DyME.
> > >
> > > There are two key reasons why prompt rewriting benefits from a “partner” like DyME:
> > > - Prompt rewriting is inherently semantic-changing.
> > > - Prompt rewriting alone cannot guarantee that a specific concept will not be generated.
> > >
> > > For example, if a user requests “a birthday card with a cute mouse character wearing red shorts and big round ears”, an aggressive prompt rewrite that removes or replaces visual cues (e.g., “... card with a generic animal”) may avoid Mickey-like outputs, but also drifts away from the user’s intended content and design constraints. On the other hand, a milder rewrite (e.g., “a birthday card with a cute mouse character”) may still cause DM to generate Mickey-like copyrighted content.
> > >
> > > DyME operates at the level of the model’s parameters and can suppress the copyrighted “Mickey-like” concept under arbitrary prompts. This allows users to retain maximum desired semantics while the model prevents the copyrighted realization.
> > >
> > >
> > > > **Q2.** The nature of concept erasure and the model bypass problem.
> > >
> > > **A2.** We agree with the reviewer that concept erasure should primarily be driven by the model provider or by regulatory requirements. Importantly, however, these safety and policy requirements are diverse and evolving across jurisdictions, age groups, and application contexts.
> > >
> > > For instance, age- and content-based rating systems such as the MPA’s G / PG / PG-13 / R categories, or similar schemes in other regions, impose different expectations on which kinds of content are acceptable for different audiences.
> > >
> > > Likewise, copyright constraints also differ: some characters or artwork may be protected in certain regions but not in others. For example, the original Winnie-the-Pooh entered the public domain in the US in 2022, while it remains under copyright in the UK until much later.
> > >
> > > Under this reality, one static erasure model cannot satisfy all use cases and regulatory regimes simultaneously. The notion of which concepts must be suppressed is inherently dynamic and context-dependent. Our framework is designed precisely for this setting: DyME allows a provider to maintain a large library of concept-specific erasure modules and to reconfigure which subset is active for a given deployment profile.  We will clarify that DyME is not intended to give arbitrary end users free control over safety policy; instead, it gives providers a scalable and modular way to implement diverse, evolving safety and compliance requirements across deployments.
> > >
> > > We also agree that in fully open-source, white-box settings, adversarial users can bypass any safety modules, including DyME, by simply loading the original checkpoint (and then no parameter-level safety method prevents this). We will explicitly state that our claims and DyME is not for defending against fully adversarial model owners.
> > >
> > > > **Q3.** For cloud providers, are prompt filters (pre-generation) and image filters (post-generation) already sufficient?
> > >
> > > **A3.** Prompt filters and image filters are indeed widely used in practice, but there are several things they fundamentally cannot do.
> > >
> > > 1. Prompt filters: Any keyword-based rule set is finite. If a user describes a concept using alternative, partial or creative phrasing that is not on the blocklist, the prompt filter simply does nothing. As in the birthday card example from Answer A1, a request might not match any explicit blocked token, yet the original SD model is still likely to produce an image with the copyrighted character.
> > >
> > > 2. Image filters: If a provider relies only on post-generation filtering and frequently replaces those borderline generations with blank or noisy images, users may experience this as confusing or frustrating. For example, when a user issues the above birthday card prompt but their outputs keep being blocked, they may feel: “I did not ask for anything inappropriate; why do I keep getting blank outputs?”
> > >
> > > Also, we want to clarify that DyME itself is not inherently based on prompt keyword matching. In our formulation, per generation we simply assume that the provider knows which subset of concept-specific LoRAs from the erasure scope should be composed. It does not require a direct “keyword to LoRA” trigger as in the prompt filter. In the above example, a provider could compose all LoRAs related to a copyrighted mouse-character concept, so that any prompt in that deployment avoids producing copyrighted outputs regardless of keywords.

---

### Official Review · Reviewer_akRp · 2025-10-30

**Soundness:** 1
**Presentation:** 3
**Contribution:** 2
**Rating:** 2
**Confidence:** 4

**Summary:**

The paper investigates dynamic multi-concept erasure by pretraining LoRA adapters and composing the required concepts at inference. To mitigate crosstalk among LoRA modules, the authors propose Bi-Level Orthogonal Constraints that minimize overlap in LoRA-induced representation shifts and enforce orthogonality in the parameter space. Experimental evaluation employs a CLIP-based classifier to assess both erasure effectiveness and utility preservation. The results indicate that the proposed method maintains the original utility while achieving the lowest accuracy for the erased concepts.

**Strengths:**

1. Unlike existing concept erasure methods, this paper focuses on dynamic, on-demand multi-concept erasure, enabling separate LoRA adapters to be combined dynamically at inference time.
2. The paper is well-structured and clearly written. The proposed DyME can mitigate LoRA crosstalk.

**Weaknesses:**

1. For the motivation, this paper addresses the dynamic combination of trained concept-specific LoRA modules to enable flexible concept erasure in diffusion models. It specifically tackles scenarios where certain concepts may need to be removed from existing LoRA sets. However, the proposed method is limited in that it can add one new concept at a time, potentially requiring retraining across all LoRA modules. The approach’s scalability is restricted if it cannot seamlessly incorporate new concepts without retraining.

2. For the evaluation metrics, the authors adopt a CLIP-based classifier to assess erasure effectiveness and utility preservation. However, existing works typically employ the CLIP score for erasure effectiveness and FID for utility preservation. It remains unclear what advantages the CLIP-based classifier provides over these established metrics. Moreover, LoRA-based weight modifications inevitably affect untargeted concept generation, even if only minimally. Yet, the results measured by $Acc_{UP}$ suggest that the method fully preserves the original performance on untargeted concepts. Therefore, $Acc_{UP}$ alone may not sufficiently capture potential degradation in untargeted concept generation.

3. As the number of erased concepts increases, the difficulty of training LoRA correspondingly rises. Therefore, it remains unclear whether the proposed method can still perform effectively when dealing with a larger set of concepts to erase.

**Questions:**

1. How flexible is the system in incorporating new concepts after initial training?
2. Can the authors clarify whether DyME remains effective when erasing a large number of concepts?
3. What is the rationale for choosing this particular evaluation metric?

---

> ### Author Response · Authors · 2025-11-25
> **Response to Reviewer #akRp**
>
> We thank the reviewer for the careful reading, the positive assessment of the practical importance of DyME. Below we address each of your concerns in turn.
>
> > **W1.**:  Practicality when users erase new concepts
>
> **A1.** We thank the reviewer for highlighting this real-world scenario. Supporting incremental addition of erasing new concepts is a central motivation of our design. Traditional static erasure methods require full model retraining whenever the erasure scope expands, which is expensive and often infeasible. Our dynamic scheme only trains lightweight LoRA adapters for the new concepts. This significantly reduces the computational cost. In our experiment on 20 new concepts, the training time required by our approach is 8.64 hours. This is substantially lower than prior methods such as SPM which requires 24.6 hours and ESD which requires 14 hours.
>
> > **W2.**:  Evaluation metric.
>
> **A2.** Thank you for raising this question about the evaluation metric. We would like to clarify that to make a fair comparison, we follow the metrics reported in existing works and use the CLIP-based accuracy and FID metrics (e.g., Table 1) in the paper (SalUn, MACE, GLoCE).
>
> We would also like to clarify that CLIP score and CLIP-based accuracy are closely related. CLIP score measures the similarity between an image and a single target concept, while CLIP-based accuracy checks whether the target concept has the highest CLIP logit among all candidate concepts, thus capturing its relative position in the label space.
>
> Additionally, there is another way CLIP score is used in existing works (e.g., ESD, MACE): they generate images conditioned on the COCO-30k label space and compute the CLIP score between each COCO label and the corresponding image. This evaluation is closer in spirit to FID, as it mainly reflects overall image–text consistency and visual realism, rather than measuring whether a specific concept has been erased or preserved.
>
> Regarding utility preservation on untargeted concepts, DyME is able to maintain the original performance because the LoRA modules can be fully deactivated when generating benign content. This plug and play property is fundamental to our design. Since the base model remains unchanged and no LoRA is active during untargeted concept generation, the output distribution for untargeted concepts is identical to that of the original model. We will clarify this design principle in the revision.
>
> > **W3.**:  Effectiveness when erasing a large number of concepts.
>
> **A3.** We appreciate the reviewer’s concern about scalability. We have conducted additional evaluations where the number of erased or composed concepts is very large.
>
> First, regarding erasure scope scaling, we extend the erasure scope from 20 to 50 concepts on CIFAR-100, as suggested by the reviewer. At this size, all baselines collapse sharply even when the erasure subset size is only 1, which means each per-generation prompt involves only one erased concept. For this evaluation, we treat the first five CIFAR-100 concepts from erasure scope, construct prompts that include each concept in turn, generate images, and compute accuracy for each class; we then report the average harmonic accuracy (Acc_harmonic) in the table below.
>
> | Method | Average Acc_harmonic ↑ |
> |--------|------------------------|
> | ESD    | 6.11%                  |
> | AC     | 16.56%                 |
> | FMN    | 24.58%                 |
> | SPM    | 40.02%                 |
> | SalUn  | 16.78%                 |
> | MACE   | 19.73%                 |
> | GLoCE  | 36.04%                 |
> | CoGFD  | 25.93%                 |
> | **DyME**   | **90.82%**                 |
>
> We also added a new Celebrities Erasure experiment where concept erasure size is 100 and erased erasure subset still involves only 1 celebrity. Still we test the first five celebrities from erasure scope, construct prompts that include each celebrity and then report the average harmonic accuracy (Acc_harmonic) in the table below.DyME again maintains stable performance (95.21 percent), outperforming all baselines by a substantial margin.
>
> | Method | Average Acc_harmonic ↑ |
> |-------|------------------------|
> | ESD   | 22.73%                 |
> | AC    | 20.19%                 |
> | FMN   | 39.92%                 |
> | MACE  | 85.95%                 |
> | GLoCE | 94.68%                 |
> | CoGFD | 41.70%                 |
> | **DyME**  | **95.21%**                |
>
> Second, regarding erasure subset scaling, Our high level prompts, such as “Mickey Clubhouse” or “Disney characters” implicitly involve up to 15 and 62 unit character concepts respectively. This requires composing a large number of adapters simultaneously. DyME consistently outperforms baselines in this challenging setting, as shown in Table 2 and Appendix Tables 7 and 8.
>
> Taken together, these results demonstrate that DyME scales effectively both when the erasure scope set grows large and when generation requires combining many concept adapters.

---

### Official Review · Reviewer_HdpA · 2025-11-06

**Soundness:** 3
**Presentation:** 3
**Contribution:** 2
**Rating:** 2
**Confidence:** 4

**Summary:**

The paper presents DYME, a modular and scalable framework for dynamic multi, concept erasure in text, to, image diffusion models. Instead of retraining or editing a model for each new erasure request, DYME trains one lightweight LoRA adapter per concept and dynamically composes relevant adapters during inference. To handle interference among multiple adapters, the authors introduce a bi, level orthogonality mechanism consisting of (1) an input, aware orthogonality constraint on induced representation shifts and (2) an input, agnostic parameter, space regularizer derived from a theoretical sufficient condition (Theorem 1). The paper also contributes ERASUREBENCH, H, a hierarchical benchmark organized into brand–series–character levels. Experiments on CIFAR, 100, Imagenette, and ERASUREBENCH, H demonstrate improved erasure efficacy and reduced interference compared to prior static erasure baselines.

**Strengths:**

1. Addresses real, world dynamic takedown scenarios where multiple concept erasures must coexist flexibly.
2. Per, concept LoRA adapters make updates lightweight and composable.
3. Provides a theoretically backed mechanism to reduce interference among multiple LoRAs.
4. ERASUREBENCH, H offers a structured and hierarchical testbed for compositional erasure.
5. Empirical results: Demonstrate consistent gains across datasets and include several informative ablations.

**Weaknesses:**

1. Missing comparison to contemporaneous SOTA (e.g., Receler).
The work omits comparisons with key 2025 approaches that address similar problems. This omission weakens claims of novelty and superiority. A direct experimental or analytical comparison with Receler and other strong 2025 methods should be added.

2. Theorem 1 vs implementation inconsistency.
The theorem assumes q and k are frozen while LoRA updates only v/o, yet the implementation adapts q/k/v/o. The authors should either align the implementation with the theorem’s assumptions or extend the theoretical justification accordingly.

3. Scalability and cost reporting.
With hundreds of unit concepts, training one LoRA per concept can be computationally expensive. The paper should report adapter size, total storage, per, adapter training time, and inference latency when composing many adapters.

4. Baseline fairness.
Static baselines were restricted to small erasure scopes due to collapse; stronger tuning or staged training might alleviate this. A justification or expanded baseline study is required.

**Questions:**

Questions*
1. Why is Receler (and other 2025 SOTA methods) not compared? If unavailable, can you provide a reasoned analysis or reimplementation?
2. Do you freeze q/k during LoRA training? If not, how does Theorem 1 apply in practice?
3. What are the per, adapter parameter counts, total storage, training time, and inference latency at different composition sizes?
4. How were static baselines tuned, and are there hyperparameter settings that allow them to scale further?

---

> ### Author Response · Authors · 2025-11-25
> **Response to Reviewer #HdpA06**
>
> We thank the reviewer for the careful reading and for recognizing that DyME addresses real-world dynamic takedown scenarios with lightweight, composable per-concept LoRA adapters, a theoretically grounded bi-level orthogonality mechanism, and the ERASUREBENCH-H benchmark, as well as its consistent empirical gains over static erasure baselines. We address each of your concerns below.
>
> > **W1.**: Missing comparison to contemporaneous SOTA methods.
>
> **A1.** Thanks for suggesting a comparison with SOTA approaches. In the revision, we have compared our proposed method with two recent SOTA methods: CoGFD (ICLR 2025) and GLoCE (CVPR 2025). CoGFD performs concept combination erasure using an LLM-induced logical graph, and GLoCE performs training-free localized erasure based on SVD-derived low-rank updates.
> We evaluated both methods under the erasure scope scaling task in Section 5.2. When the erasure scope is 50 concepts, the harmonic accuracy of CoGFD and GLoCE drops to 25.92 percent and 36.04 percent respectively, while our proposed DyME achieves 90.82 percent under the same setting.
>
> The performance gap arises because neither CoGFD nor GLoCE is designed for large-scale multi-concept erasure. CoGFD focuses on a small set of high-level combinations, and its logic graph does not scale to dozens or hundreds of unit concepts whose combinations grow combinatorially. GLoCE is effective for single or a few concept erasure, but it composes multiple gated low-rank modules without mechanisms to prevent cross-concept interference. When multiple gates are activated simultaneously, their updates accumulate in shared feature spaces, degrading performance. DyME directly addresses these scalability issues, which explains its advantage in large erasure scopes.
>
> > **W2.**: Theorem versus implementation inconsistency for q and k.
>
> **A2.** Thank you for pointing out this detail. In all main experiments reported in Sections 5.2 and 5.3 and in Tables 1, 2, 3, 6, 7, 8, and 9, we strictly follow the theoretical assumptions and freeze both q and k during LoRA training.
>
> We allow q and k to be trainable only in the ablation study in Section 5.4 (Table 4). The goal is to evaluate the input agnostic constraint derived in Theorem 1. The ablation results show that relaxing this constraint by training q and k leads to degraded erasure performance, confirming the importance of our theoretical insight. We will clarify this distinction in the revision.
>
> > **W3.**: Scalability and cost.
>
> **A3.** We provide the detailed cost breakdown below, measured on an NVIDIA A100. The cost is from the erasure scaling experiments on CIFAR-100 concepts. The erasure scope is set to 20 and the erasure subset size is 1. Acc_harmonic are the overall performace of models. Image generation latency is measured per generation with 50 DDIM steps.
>
> | Method | Data Preparation (hour) | Model FineTuning (hour) | Image Generation latency (second) | Average Acc_harmonic↑ |
> |-------|-----------------|---------------|----------------|------------------------|
> | ESD   | 0               | 14            | 3              | 9.92                   |
> | AC    | 3               | 5             | 3              | 13.50                  |
> | SPM   | 0               | 24            | 6              | 65.48                  |
> | MACE  | 0.46            | 0.4           | 3              | 25.43                  |
> | Ours  | 0.46            | 8.4           | 3.1           | 90.82                  |
>
> Key metrics for our method:
> - Data preparation time: 0.46 hours
> - LoRA fine-tuning time: 8.4 hours
> - Image generation latency: 3.1 seconds per image
> - Adapter size: approximately 6.2 MB per concept
>
> The results show that our method is lightweight while achieving the best erasure performance, and both training and inference latency remain practical even when multiple adapters are composed.
>
> > **W4.**: Fair comparison with baselines.
>
> **A4.** We clarify that all baselines are trained using the official configurations, including optimizer settings, learning rates, LoRA ranks and scales, training schedules, loss functions, and early stopping rules. For methods that require concept mappings, we use GPT-4 to generate mappings.
>
> All baselines are trained to convergence, as the figure ([link](https://anonymous.4open.science/r/ForAnonymousGithub-C9D0/TrainingLossCurve.png)) shows. Each model erases 20 concepts; the bold curve denotes the mean training loss across the 20 concepts, and the surrounding region represents the range between the min and max loss over these concepts. Their normalized losses all trigger early stopping (for example, MACE at 1e-4, SPM at 1e-4), indicating that training instability or insufficient tuning is not the main cause of performance collapse. Instead, the collapse arises from the structural limitations of static methods as the erasure scope expands. This aligns with the findings reported in their original papers. We will add a short discussion to clarify this point.

---

### Note · Authors · 2026-01-07

I have read and agree with the venue's withdrawal policy on behalf of myself and my co-authors.